# Compositional Generalization via Forced Rendering of Disentangled Latents

Qiyao Liang* [1]   Daoyuan Qian* [2]   Liu Ziyin [1][3]   Ila Fiete [1]

## Abstract

Composition—the ability to generate myriad variations from finite means—is believed to underlie powerful generalization. However, compositional generalization remains a key challenge for deep learning. A widely held assumption is that learning *disentangled* (factorized) representations naturally supports this kind of extrapolation. Yet, empirical results are mixed, with many generative models failing to recognize and compose factors to generate out-of-distribution (OOD) samples. In this work, we investigate a controlled 2D Gaussian "bump" generation task with fully disentangled $(x, y)$ inputs, demonstrating that standard generative architectures still fail in OOD regions when training with partial data, by re-entangling latent representations in subsequent layers. By examining the model's learned kernels and manifold geometry, we show that this failure reflects a "memorization" strategy for generation via data superposition rather than via composition of the true factorized features. We show that when models are forced—through architectural modifications with regularization or curated training data—to render the disentangled latents into the full-dimensional representational (pixel) space, they can be highly data-efficient and effective at composing in OOD regions. These findings underscore that disentangled latents in an abstract representation are insufficient and show that if models can represent disentangled factors directly in the output representational space, it can achieve robust compositional generalization.

## 1. Introduction

A core motivation for disentangled representation learning is the belief that factoring an environment into independent latent dimensions, or "concepts", should lead to powerful combinatorial generalization: if a model can learn the representation of each factor separately, then in principle it can synthesize new combinations of those factors without extensive retraining. This property is often referred to as *compositional generalization* and is considered crucial for data efficiency and robust extrapolation in high-dimensional tasks, such as language understanding and vision (Lake & Baroni, 2018).

Despite many attempts, disentangled representation learning has shown mixed results when it comes to compositional generalization. Some studies report clear data-efficiency gains and OOD benefits (Higgins et al., 2017a; Chen et al., 2018), while others find no significant advantage or even detrimental effects (Locatello et al., 2019; Ganin et al., 2017). This tension motivates a deeper theoretical exploration:

> *Is a disentangled/factorized[1] representation alone sufficient for compositional generalization? If not, why, and what other ingredients or constraints are necessary?*

In this paper, we develop a mechanistic perspective on why even fully disentangling the intermediate (or input) representations is often insufficient to enable robust out-of-distribution (OOD) generalization. Concretely, we focus on a synthetic 2D Gaussian "bump" generation task, where a network learns to decode given $(x, y)$ coordinates into a spatial image. We provide a detailed empirical investigation into why *disentangled representations* often fail to achieve robust compositional generalization. Specifically, we show that:

- From a representational manifold viewpoint, even if the input or bottleneck (latent) layer are fully disentangled, subsequent layers usually "warp" and remix factors, leading to poor out-of-distribution (OOD) extrapolation.
- From a kernel-based perspective, networks failing to

---

[1] We use the terms "disentanglement" and "factorization" interchangeably. See Section A.2 for a discussion of the definitions and distinctions.

*Equal contribution [1]Massachusetts Institute of Technology, Cambridge MA, USA 02139 [2]University of Cambridge, Cambridge CB2 1EW, U.K [3]NTT Research. Correspondence to: Qiyao Liang <qiyao@mit.edu>.

*Proceedings of the $42^{nd}$ International Conference on Machine Learning*, Vancouver, Canada. PMLR 267, 2025. Copyright 2025 by the author(s).

compose have overwritten their disentangled inputs by "memorizing" training data, rather than combining independent factors; they simply superpose memorized states when attempting OOD generalization.

- Using a Jacobian-based metric tensor to study the representational manifold, there is a layer-by-layer erosion of network disentanglement in standard CNN decoders.
- Finally, forcing disentangled inputs to render into a representation that matches the output (pixel) space — via architectural constraints or curated datasets—enables the model to overcome memorization strategies and learn genuinely compositional rules. This approach leads to data-efficient, out-of-distribution generalization in cases where factorized latents alone fail.

In addition, we provide in Appendix A a general framework for interpreting compositionality and factorization that further clarifies these observations. Overall, these results caution against the assumption that a factorized bottleneck automatically confers compositional extrapolation, and they point toward more comprehensive design strategies, such as disentangled processing and data curriculum that enforce the rendering of disentangled latents, to achieve genuinely compositional neural models. [2]

## 2. Related Work

**Disentangled representation learning.** There is an extensive body of work on disentanglement, ranging from early theoretical proposals (Bengio et al., 2013) to various VAE-based approaches (e.g. $\beta$-VAE (Higgins et al., 2017a), FactorVAE (Kim & Mnih, 2018)) and GAN-based methods (Chen et al., 2016). Empirical metrics to quantify disentanglement include FactorVAE score, MIG score, and others (Eastwood & Williams, 2018). Yet, these often focus on axis-aligned or linear independence in the latent space, without explicitly evaluating compositional out-of-distribution performance. Indeed, (Locatello et al., 2019) show that unsupervised disentanglement is sensitive to inductive biases and data assumptions, raising questions about the connection to systematic compositional generalization.

**Compositional generalization & disentanglement.** Factorization and compositional generalization have been widely investigated in various architectures (Zhao et al., 2018; Higgins et al., 2017b; Burgess et al., 2018; Montero et al., 2021; Xu et al., 2022; Okawa et al., 2023; Wiedemer et al., 2023; Lippl & Stachenfeld, 2024; Liang et al., 2024b). While many works study how $\beta$-VAEs and related methods can learn disentangled representations (Burgess et al., 2018; Higgins et al., 2017a), they typically do not explore whether these representations solve the problem of compositional extrapolation—particularly in the presence

of mixed discrete and continuous features. However, Xu et al. (2022) introduced an evaluation protocol to assess compositional generalization in unsupervised representation learning, discovering that even well-disentangled representations do not guarantee improved out-of-distribution (OOD) behavior. Similarly, Montero et al. (2021; 2022) concluded that a model's capacity to compose novel factor combinations can be largely decoupled from its degree of disentanglement; and (Lippl & Stachenfeld, 2024) derived and demonstrated numerous failure modes in compositional generalization with disentangled latents. These observations temper the once-common assumption that factorized latents automatically yield systematic compositional generalization. Consequently, a gap remains in understanding how—and to what extent—models that appear to disentangle latent factors can also robustly compose them under large distribution shifts, and how to encourage such generalization. Our work aims to rigorously illuminate this gap in the context of a controlled task that enables characterization of computational generalization performance starting from exactly disentangled latents. Generalization is tested under large distributional shifts, and we investigate the detailed mechanisms of compositionality by the deep neural network decoders, providing a complementary perspective to prior disentanglement benchmarks and broader generative modeling approaches.

## 3. Why Disentangled Latents Often Fail Compositional Generalization

We present a toy 2D Gaussian "bump" generation task to highlight and analyze a core phenomenon: even with explicitly disentangled inputs or bottleneck, standard feedforward networks can fail to generalize compositionally.

### 3.1. Toy example: 2D Gaussian bump generation

**Task setup.** We consider a generative model that must output an $N \times N$ grayscale image containing a Gaussian "bump" at a specified 2D $(x, y)$ location within some bounding box (e.g. $[0, N]^2$). The training set covers a square-donut shaped region of $(x, y)$ with a large OOD region in the center of the training distribution (Fig. 1(a)); alternatively, the OOD region is an equivalent area in the bottom left corner of image. In both cases, the model sees all $x$ and $y$ values in training but many combinations are held out.

**Architecture.** To distinguish the behavior of the decoder from the encoding process that must learn (and may fail to fully disentangle) its latent representation, we explicitly construct a fully disentangled latent representation and ask whether the downstream decoder can leverage that factorization for compositional generalization. Concretely, we focus on a CNN-based "decoder-only" architecture that maps disentangled latent inputs to image outputs. We also

---

[2]Code available at github.com/qiyaoliang/DisentangledCompGen

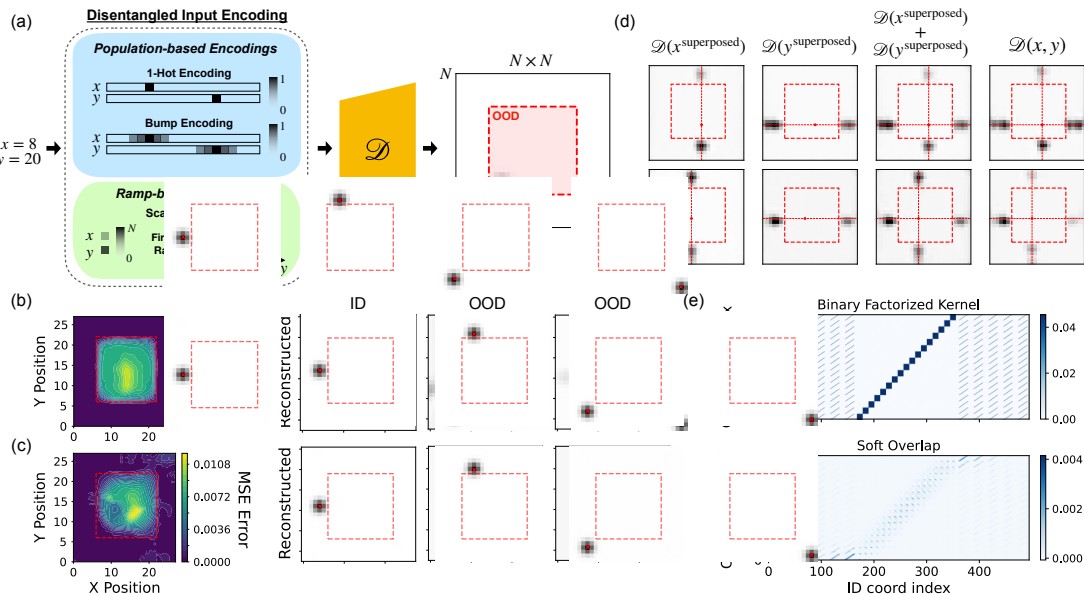

*Figure 1.* **Various disentangled input/latent encodings fail to support compositional generalization.** (a) shows the experimental setup for the 2D Gaussian bump toy experiment, where the scalar $(x, y)$ coordinate pair is encoded into disentangled representation (population-based vs. ramp-based encodings), which is then fed to a decoder-only architecture to generate a $N \times N$ grayscale image of a single 2D Gaussian bump centered at that corresponding $(x, y)$ location. The training dataset excludes all images that contain 2D Gaussian bumps centered within the red-shaded OOD region in the center of the image field. (b-c) shows the MSE error contour plots and sample generated ID/OOD images of bump-based and ramp-based encoding of the $x$ and $y$ input, the compositional OOD region is marked with by the red-dashed bounding box and the ground truth bump location is marked by a red cross. For a non-compositional network trained with bump-encoded inputs, (d) demonstrates that it learns to "superpose" seen ID training data when asked to compositionally generalize, and (e) shows the agreement between a theoretical binary factorized kernel with the similarity matrix (computed based on the pixel overlap) between the model's ID and OOD generated samples.

experiment with MLP decoders and provide those results in Appendix F.3, observing qualitatively similar findings.

**Disentangled input encodings.** Previous work suggests that vector-based representations may be key to compositional generalization (Yang et al., 2023). Inspired by stimulus coding within brains (Georgopoulos et al., 1986; O'Keefe & Nadel, 1978; Shadlen & Newsome, 1998), we test various input encodings:

1. *Population-based coding*, e.g. a 1-hot, local-bump or positional encoding of $x$ and $y$,
2. *Ramp coding*, an analog ramping code for $(x, y)$ in 2 units.

In each case, the input representation is disentangled w.r.t. $x$ and $y$ (meaning $x$ and $y$ are independently encoded by separate neurons). These inputs drive an image-generating CNN.

### 3.2. Model fails to compositionally generalize across various disentangled encodings.

Despite receiving disentangled inputs that correctly specify OOD $(x, y)$ coordinate combinations, the model *does not*

learn to generate the correct outputs beyond the ID region, as seen in the MSE error landscapes of generated images and from the images themselves Fig. 1(b-c). Networks trained with different input encodings, bumps ($x$ and $y$ inputs embedded as the centers of 1D Gaussian bumps in two vectors of length $N$, Fig. 1(b), right) or ramps ($x$ and $y$ represented as analog ramping rates in two units, Fig. 1(c), right) demonstrate different degrees of "local" generalization: some input encodings can support bumps partially inside the OOD region if they overlap with the ID region). However, all networks largely fail in the OOD region, and all consistently generate multiple bumps when prompted with an OOD $(x, y)$ input.

Hence, simply providing factorized inputs, regardless of the form of encodings, does not suffice. The deeper layers appear to re-entangle $x$ and $y$ or rely on memorized local features, leading to poor compositional generalization. While not shown, models trained with disentangled 1-hot and positional input encodings also fail to compositionally generalize. These findings confirm that disentangled latents, across a variety of encoding formats, are not sufficient for compositional generalization.

For the remainder of the manuscript, we use inputs encoded as local bumps or ramps.

### 3.3. Model "superposes" seen data when asked to generalize.

The form of failure in OOD generation is suggestive: In many runs, the CNN appears to memorize the training distribution and when prompted with an OOD $(x, y)$ combination, seems to perform a lookup: it finds an image or images in the training data with similar values of $x$ and the closest values of $y$, and superposes these to generate a new image, Fig. 1(b-c), right.

The consistency of these patterns across input encoding formats and architectures indicate a shared underlying mechanism of generalization that we further characterize here.

**Kernel-based approach to characterizing OOD generalization.** One approach to characterizing model's performance on unseen data is to characterize a kernel $K$ : $\mathcal{Z} \times \mathcal{Z} \to \mathbb{R}$, where $\mathcal{Z} = \mathcal{X} \times \mathcal{Y}$ is the cartesian product input domain of $x$ and $y$ coordinates of the Gaussian bumps in our toy example. Here we will focus our discussion to the 2-feature scenario for 2D Gaussian bump generation. A generalized introduction to kernel-based approach in characterizing factorization and OOD generalization is given in the Appendix A.4. We note that any positive-semidefinite, symmetric, and square-integrable kernel $K$ can be decomposed into a Schmidt decomposition (for the 2-feature case):

$$K\big((x, y), (x', y')\big) = \sum_{r=1}^{\infty} \lambda_r \, k_x^{(r)}(x, x') \, k_y^{(r)}(y, y'), \quad (1)$$

where each term $k_x^{(r)}(x, x') \, k_y^{(r)}(y, y')$ is *rank-1* with respect to the $(x)$ and $(y)$ factors, and $\{\lambda_r\}$ are nonnegative eigenvalues. The number of terms with nonzero $\lambda_k$ determines how mixed the kernel is. In practice, we can construct the discrete approximation, the *Gram matrix* $\mathbf{K}$ on a finite set of data inputs $\{(x^{(i)}, y^{(i)})\}_{i=1}^{M}$ and their corresponding outputs:

$$\mathbf{K}_{(i,j)} = K\Big((x^{(i)}, y^{(i)}), (x^{(j)}, y^{(j)})\Big). \quad (2)$$

**Non-compositional models learn a binary factorized kernel.** In the kernel language, we would expect a compositional model to break down a novel input combination $(x, y)_{\text{OOD}}$ into factorized kernel, each independently mapping $x$ and $y$ to the seen components via $K_x$ and $K_y$. From the generated OOD outputs of the networks shown in Fig. 1(b-c), we observe that the generated bumps seem to align with the ground truth $x$ and $y$ coordinates of the OOD bump. Empirically, we found that this is due to the model

memorizing the ID data and "superposing" activations corresponding all or some of the seen data of $x$ and $y$ when asked to generalize to $(x, y)_{\text{OOD}}$.

In Fig. 1(d), we show that the model's output on a novel input pair $\mathcal{D}((x, y)_{\text{OOD}})$ is the combination of all of the model outputs with the "superposed" inputs, $x^{\text{superposed}} = (x, \sum_i^{M_{\text{ID}}} y^{(i)} / M_{\text{ID}})$ and similarly $y^{\text{superposed}} = (\sum_i^{M_{\text{ID}}} x^{(i)} / M_{\text{ID}}, y)$, i.e. $\mathcal{D}((x, y)_{\text{OOD}}) \approx \mathcal{D}(x^{\text{superposed}}) + \mathcal{D}(y^{\text{superposed}})$. Here $M_{\text{ID}}$ is the number of ID data. To be more concrete, in the language of kernel, we can define *a binary factorized kernel*,

$$K_{\text{binary}}\big((x_i, y_i), (x_j, y_j)\big) = \kappa_x\big(x_i, x_j\big) \otimes_{\text{OR}} \kappa_y\big(y_i, y_j\big) \quad (3)$$

$$= \kappa_x\big(x_i, x_j\big) + \kappa_y\big(y_i, y_j\big) - \kappa_x\big(x_i, x_j\big) \kappa_y\big(y_i, y_j\big), \quad (4)$$

where $\kappa_x(x_i, x_j) = \kappa_y(y_i, y_j) = \delta_{i,j}$ is given by the Kronecker delta function. With normalization, the Gram matrix element is given by $\tilde{K}_{i,j} = \frac{K_{\text{binary}}(i,j)}{\sum_{j'} K_{\text{binary}}(i,j')}$. Fig. 1(e), we visualize the similarity between representations of OOD-generated samples (sorted by coordinates) and an idealized binary kernel, where similarity equals 1 if coordinates match exactly and 0 otherwise. Specifically, Fig.1(e) compares the ideal binary kernel (top panel) to the actual similarity matrix derived from model-generated Gaussian bump images (bottom panel). The *agreement* between these two matrices quantifies how closely the model's learned similarity structure aligns with this ideal binary factorized kernel. Fig. 1(e) corresponds specifically to the top left sections of the full matrices shown in Fig.8 in Appendix F.1, which provide a comprehensive mapping of similarities between all ID and OOD generated samples. These results indicate that the disentangled input structure enables the model to independently map the $x$ and $y$ coordinates to in-distribution (ID) samples, a form of novel OOD generation that is not the same as OOD compositional generalization.

In a set of follow-up experiments on MNIST image rotation, we observe that this same superposition of memorized activation patterns—drawn from related in-distribution data—emerges as a general OOD generalization strategy in neural networks that have learned to memorize ID data. Crucially, this phenomenon applies not only to compositional tasks but also to single-dimension interpolation and extrapolation. We detail these results in Appendix F.2.

Similarly, in CNN-based guided diffusion models tasked with Gaussian bump image generation, (Liang et al., 2024a) reported that the model generated a superposition of multiple ID bumps when guided to generate an OOD bump. Similar to both these findings, (Kamb & Ganguli, 2024) showed that guided diffusion models generate new images by gluing together local patches from the closest relevant images in the training data. Together with (Lippl & Stachenfeld, 2024), these observations suggest an emerging consen-

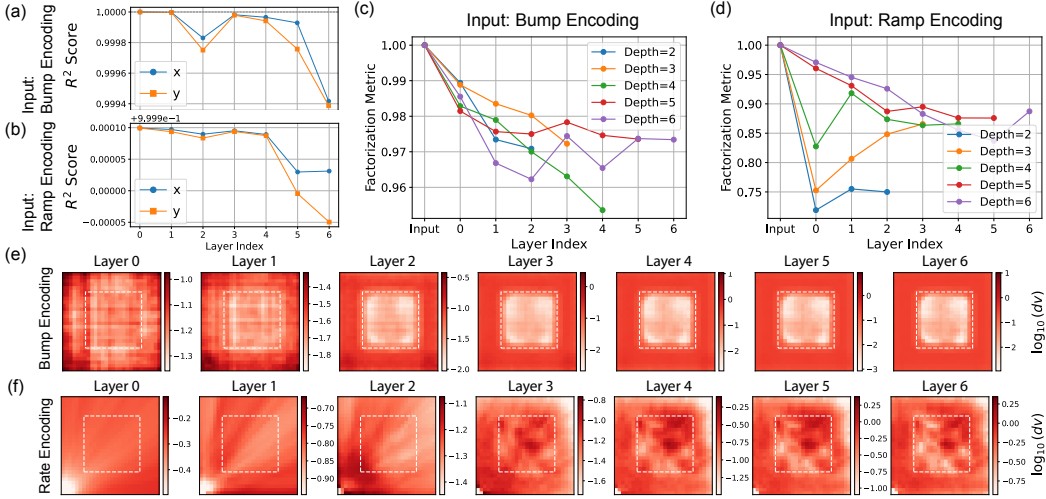

*Figure 2.* **Cause of failure of compositional generalization despite disentangled inputs/latents: input memorization by decoder undoes factorization.** (a-b) show the linear probe metrics of the learned representation as a function of $x$ and $y$ for bump- and ramp-input-encodings, and (c-d) show the factorization score (Eq. 7), and (e-f) show the volume metric $\log_{10}(dv)$, all as a function of layer depth, for models trained with bump-based vs. ramp-based input encodings. The linear probe metric is defined as the $R^2$ scores of fitting two linear classifiers with respect to $x$ and $y$.

sus view of how various generative deep networks function in OOD generation – by combining literal patches of their training data, rather than combining factors from the data, even when provided with information on what these factors should be in the form of disentangled inputs.

### 3.4. Manifold warping through dimension expansion ruins factorization in latent space.

Why is disentangled input insufficient for learning disentangled factors of variation? Here we seek deeper mechanistic insight into model's learned representations along the network layers. Specifically, we track how strongly $x$ and $y$ remain disentangled across layers. While the *input* layer is disentangled (by construction), subsequent convolutional layers produce *mixed* features with large cross-terms, particularly near the final decoding layer. This "manifold warping" effectively destroys compositional structure, leaving the network reliant on patterns that do not extend to unseen $(x, y)$ combinations.

**A transport perspective.** From a transport perspective, the model layers effectively learn to interpolate between the input (source) distribution and the output (target) distribution. Here the source distribution corresponds to the disentangled input representation manifold, while the output corresponds to the ID data representation manifold, which has an OOD "hole" in the middle. If there is nontrivial topological or geometrical deformation to the source or the target distribution representations, the model will have a hard time maintaining the integrity of the factorization, especially through dimension expansion (or reduction equiv-

alently) in the generation process. Let the network be a composition of layers $T = T_L \circ \cdots \circ T_1$, with each $T_l : \mathcal{Z}_{l-1} \to \mathcal{Z}_l$. The input measure $\mu_0$ (assumed factorized) on $\mathcal{Z}_0$ is "pushed forward" through these layers to match the in-distribution measure $\mu_L^{\text{ID}}$ on the final space $\mathcal{Z}_L$. Formally, $\mu_l = (T_l \circ \cdots \circ T_1)_* \mu_0$ for $l = 1, \ldots, L$, so $\mu_L = T_* \mu_0$ should approximate $\mu_L^{\text{ID}}$. If $\mu_L^{\text{ID}}$ differs topologically or geometrically from $\mu_0$ (e.g. having an OOD "hole"), each layer $T_l$ may need to warp coordinates significantly. These warping steps can easily re-entangle originally factorized inputs, leading to failures in OOD regions.

**Characterization of manifold warping and factorization.** To qualitatively investigate the warping and its effect on the factorization of the layer-wise representations, we compute the Jacobian-based metric tensor $g$, inspired by the approach of Ref. (Zavatone-Veth et al., 2023). The generalized definition can be found in Appendix B. In short, the metric tensor is defined by

$$g(x, y) = J(x, y)^\top J(x, y), \quad (5)$$

where $J$ is the Jacobian matrix w.r.t. the different input feature dimension, which corresponds to $x$ and $y$ in our synthetic toy setting. We then visualize the volume metric

$$dv(x, y) = \sqrt{\det(g(x, y))} \, dx \, dy \quad (6)$$

for representation output at each of the network layer. The metric visualizations $\log_{10}(dv)$ are compared between networks with bump-based vs. ramp-based encodings in Fig. 2(e) and (f). Visually, we observe significant distortion as a function of network depth, especially in the OOD

region. To accompany the distortion analysis, we plot the linear probe metric (linear classifier $R^2$ w.r.t $x$ and $y$) as well as the factorization metric proposed based $g$ in Eq. (7) as a function of network depth in Fig. 2(a-d). Here the factorization metric is defined as follows

$$\text{Factorization}(g(x,y)) = 1 - \frac{\left|g_{xy}(x,y)\right|}{\left|g_{xx}(x,y)\right| + \left|g_{yy}(x,y)\right|},\tag{7}$$

where $g_{ij}$'s are the matrix elements of the $2 \times 2$ matrix $g$. We note that both metrics decay relatively gracefully over the layer depth for both networks with bump-based and ramp-based input encodings. Intriguingly, shallow networks with ramp-based encoding experiences a sharper drop in factorization. This is likely due to the higher demand for dimension expansion with the ramp-based 2-neuron input, which leads to higher distortion when allowed fewer network layers to accommodate the expansion. The volume metric visualization as well as the linear probe metric as a function of layer depth for different network depths are shown in Fig. 11 and Fig. 12 (Fig. 13 and Fig. 14 for MLPs) as well as generated OOD images in Fig. 10 in Appendix F.3.

# 4. Encouraging Compositional Generalization

Given the above model failures, we ask: *Which architectural or training strategies can preserve factorization and yield robust compositional generalization?* We provide two promising avenues next.

## 4.1. Architecture regularization for low-rank embeddings

One approach we adopt is to architecturally encourage the expansion of abstract latent representations into structured 2D embedding filters, while applying explicit rank regularization during training (Fig. 3(a)). This design promotes interpretability and compositional disentanglement in the learned features. For example, Cichocki et al. (2009) demonstrate that enforcing low-rank tensor factorization can mitigate spurious interactions between factors. Inspired by this, we apply a matrix (or tensor) factorization penalty to the set of 2D embedding filters associated with each input dimension, encouraging them to remain approximately low-rank and spatially consistent.

To formalize this, in the 1-hot input encoding setup, we generate embedding matrices of shape $N \times N$ for each token $x$ and separately for $y$ (Fig. 3(a)). There are a total of $2N$ such matrices. Denoting one of them as $W_{i,j}$, we decompose this matrix using an SVD-like form:

$$W_{i,j} = \sum_{n=1}^{r} v_i^{(n)} \lambda^{(n)} u_j^{(n)},\tag{8}$$

where $r$ is a chosen upper bound on the rank of the de-

composition, $\lambda^{(n)}$ are scalar singular values, and $v_i^{(n)}$, $u_j^{(n)}$ are orthogonal basis vectors along the row and column dimensions, respectively. This formulation enables us to directly regularize the singular value spectrum and the spatial smoothness of basis vectors, promoting low-rank structure in each embedding matrix.

This regularization is particularly effective because each input dimension is associated with a dedicated 2D embedding matrix, which constrains the model to learn spatially structured factors aligned with individual input dimensions. In practice, we observe that this encourages the model to discover localized or "stripe-like" patterns in the learned embeddings (see Fig. 3(b)), facilitating compositional generalization. To further stabilize learning and reduce redundancy, we additionally penalize the entropy and variance of the singular values, which promotes sparsity and consistent usage of the factorized components.

The squared magnitudes $[\lambda^{(n)}]^2$ define a probability distribution across $n$:

$$\widetilde{\lambda}^{(n)} = \frac{[\lambda^{(n)}]^2}{\sum_{m=1}^{r} [\lambda^{(m)}]^2},\tag{9}$$

which we regularize using an entropy penalty:

$$\mathcal{L}_{\text{ent}} = -\eta_1 \sum_{n=1}^{r} \widetilde{\lambda}^{(n)} \ln \widetilde{\lambda}^{(n)},\tag{10}$$

encouraging dominance of only a few modes to enforce low rank. Additionally, to discourage memorization of individual indices, we penalize variance in $v^{(n)}$ and $u^{(n)}$:

$$\mathcal{L}_{\text{var}} = \eta_2 \sum_{n=1}^{r} \Big[ \text{Var}(v^{(n)}) + \text{Var}(u^{(n)}) \Big].\tag{11}$$

The total loss function combines these terms with standard reconstruction loss:

$$\mathcal{L}_{\text{total}} = \mathcal{L}_{\text{MSE}} + \mathcal{L}_{\text{ent}} + \mathcal{L}_{\text{var}}.\tag{12}$$

Applying these regularizations to the embedding matrices of both $x$- and $y$-factors results in more structured and factorized representations, improving generalization to unseen coordinate pairs. Fig. 3(b) illustrates that, without regularization, the model struggles to preserve signal integrity in OOD regions. With regularization, the embeddings exhibit structured banding aligned with coordinate axes, indicating a discovered factorized representation beneficial for compositional extrapolation. In addition, Fig. 3(c) shows that our regularization works even for a corner OOD region, indicating a stronger form of (extrapolative) compositional generalization. With regularization, the learned representation is significantly smoother than without, as evidenced in Fig. 3(d), despite having the same factorized embedding architecture. These results demonstrate the effectiveness of our proposed regularization technique.

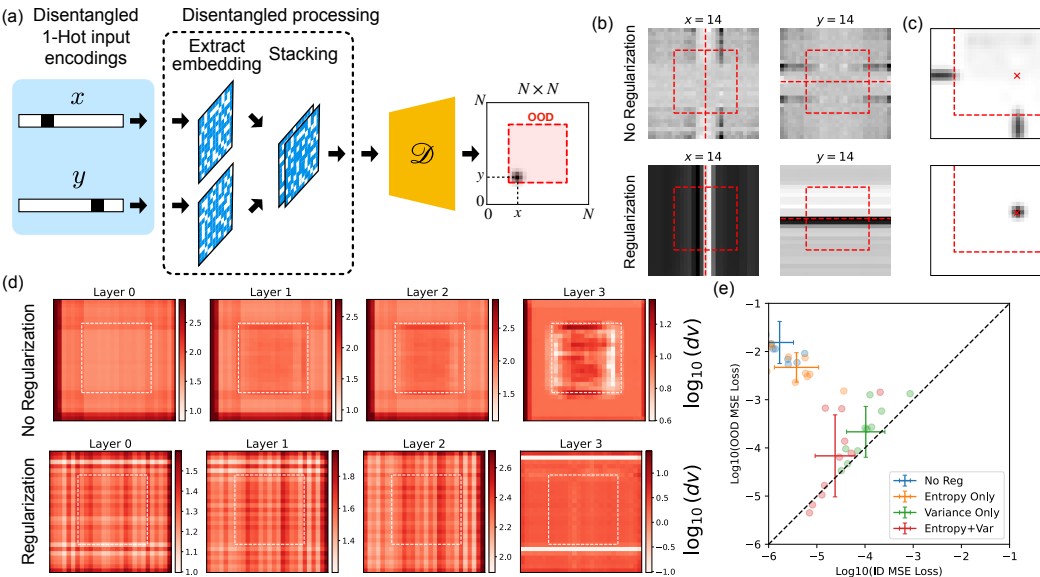

*Figure 3.* **Inducing compositional generalization through architectural rendering and regularization constraints.** (a) Schematic of architecturally forced rendering of the initially disentangled representations (with 1-hot input encodings) into a space matching the output space (disentangled processing). (b) Sampled embedding activations corresponding to $x = 14$ and $y = 14$ for networks trained without and with regularization, respectively. (c) Generated images when the OOD region is on the top right corner, for the non-regularised (top) and regularised networks (bottom) respectively. (d) Volume metric comparison between networks trained without and with regularization as a function of layer depth, respectively. (e) OOD vs. ID MSE plot for various ablation studies over many runs.

**Ablation.**    To assess the impact of our regularization strategy, we compare four configurations: (1) *No Reg* (baseline MSE loss), (2) *Entropy Only* (applying Eq. (10)), (3) *Variance Only* (applying Eq. (11)), and (4) *Entropy+Var* (combining both). Fig. 3 (e) plots MSE in distribution (ID) versus out-of-distribution (OOD). The condition *No Reg* yields the lowest ID error but the highest OOD error, reflecting poor generalization. *Entropy Only* marginally improves OOD performance, while *Variance Only* significantly enhances extrapolation at the cost of higher ID loss. Combining both constraints (*Entropy+Var*) achieves the best OOD performance while maintaining low ID error, striking the optimal balance for robust factorization.

Intuitively, the regularization leverages *simple, low-dimensional structure* in the learned embedding matrices. By limiting the effective rank, it forces the network to represent each factor (for example, a coordinate dimension) through a small set of shared directions rather than separately memorizing every possible input combination. Consequently, once the model learns an appropriate low-rank basis for in-distribution data, it can more easily *compose* those basis elements to handle new, unseen factor combinations in the OOD region. In other words, restricting the embedding to a few dominant modes discourages the network from relying on specific entries for every $(x, y)$, thus reducing overfitting and promoting a genuinely factorized representation that better generalizes to novel coordinates.

### 4.2. Dataset augmentation

Another approach that successfully enabled compositional generalization in the toy setting is dataset curation. Asking the network to generate independent factors of variation in the form of separate $x$ and $y$ "stripes" help the network discover a representation that enables compositionality.

Concretely, we generate 1D *vertical* and *horizontal* stripes by fixing a single coordinate (either $x$ or $y$) and applying a 1D Gaussian in the orthogonal direction (samples shown in Fig. 4(a), full set shown in Fig. 7(a)). Note that here the 2D input structure to the model is still maintained via setting the coordinate to the fixed dimension to $-1$. These stripes in the full output representational space act as building blocks that allow the network to learn each factor of variation *independently* in a way that abstract factorized inputs are unable to. Optionally, we additionally include a small number of the 2D bumps (outside the OOD region, as before) in the training set, so the model also sees a small number of 2D examples. Consequently, the network acquires strong *compositional* ability: it generalizes to novel $(x, y)$-combinations outside the original training distribution (even with zero 2D bumps). Empirically, as demonstrated in Fig. 4, this approach markedly improves compositional generalization performance.

**Data efficiency scaling.**    Remarkably, we observed that models trained on a dataset consisting of just the stripes

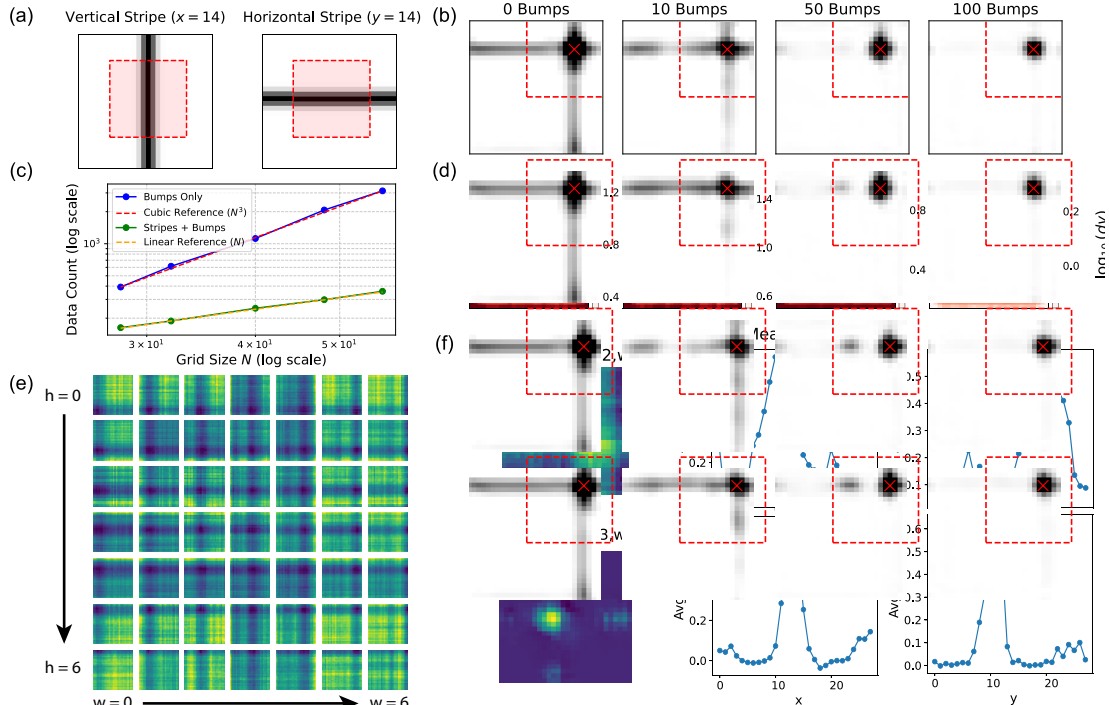

*Figure 4.* **Inducing generalizable composition by training single disentangled factors to render (data curriculum).** (a) Sample grayscale images of 1D Gaussian "stripes" at $x = 14$ and $y = 14$. (b) Generated OOD output as a function of number of Gaussian bumps (all outside the upper right OOD region) included in the training dataset. (c) Data/sample efficiency: data scaling of the stripes-only ($\sim N$) vs. bumps-only ($\sim N^3$) datasets to reach 90% accuracy of $x$ and $y$ generation as a function of image size $N$ (accuracy assessed based on the location of the darkest pixel). Here the stripes + bumps dataset consists of $\sim 7N$ bumps + $2N$ stripes; learning breaks the curse of dimensionality due to the ability to compositionally generalize zero-shot. (d) Volume metric as a function of layer depth of the network trained on a dataset of stripes + bumps. (e) 2D neuron activations across different channels in the first layer (layer 0) of a network trained trained on a dataset of stripes + 50 bumps. (f) Neural tuning curves of two sample neurons at layer 0 and layer 3 of the same network as in (e).

are able to compositionally generalize to generate additive stripe conjunctions in the OOD region without having seen any compositional examples (inside or outside the OOD region) during training, as shown in Fig. 4(b) when 0 2D Gaussian bump images are included in the training dataset. Similar to Fig. 3(c), we show here an example of corner OOD region to demonstrate that our method is robust for extrapolative compositional generalization. The remaining panels of Fig. 4(b) show that as we increase the number of 2D Gaussian bump images included in the training dataset, the model learns to generate proper bumps in the OOD region (also see Fig. 7(b)). We further characterize the data efficiency of this approach to learning to compositionally generate bumps in the OOD region (learning with stripes + bumps) *versus* a traditional approach to learning with just bumps. The total data count scaling with respect to the image size $N$ is shown in Fig. 4(c). Here we benchmark the compared models based on the minimal threshold amount of data needed to reach a target accuracy of 90% in generating the correct $x$ and $y$ coordinates of the stripe conjunc-

tions/bumps. To our surprise, we found that models trained on datasets consisting of purely bumps have a data threshold that scales as $\sim N^3$, and hence the data requirements grow as the cube of the size of the problem. On the other hand, due to the zero-shot generalization capability of models trained on the stripe dataset, the scaling is linear with $N$, since the data set consists of $2N$ stripes in total, $N$ for $x$ and $N$ for $y$ (together with 0 or a small number of bumps). Furthermore, with the increase of stripes training with bumps to obtain bump outputs rather than plus-shaped ones, the minimum required number of training bumps scales only linearly with $N$, meaning that the total training data remains linear in $N$, indicative of compositional generalization.

For comparison with the earlier results, we again plotted the volume metrics for the model trained on the stripes + bumps dataset, as a function of model layer depth, Fig. 4(d). The representations remain relatively smooth and uniform even in the OOD region, contrasting the models trained with just the bumps shown in Fig. 2. In addition, we show in Fig. 4(e) the 2D activations of neurons in the first layer of a

network trained on a dataset of stripes + 50 bumps. The activations show consistent "stripe" patterns that resemble the embeddings shown in Fig. 3(b) with regularization. Moreover, we show neural tuning curves w.r.t. to $x$ and $y$ for two sample neurons in layer 0 and layer 3 of the same network in Fig. 4(f), which exhibits "stripes" in early layers, which localizes to become a "bump" in downstream layers (more sample neural tuning curve can be found in Fig. 15).

Intuitively, the early-layer stripes allow the network to build a "scaffold" in its representations that are straightforward for it to use to perform additive composition. Later layers in the model then leverage this scaffold to handle the multiplicative compositional task — to generate stripe intersections instead of unions — thereby achieving efficient bump generalization in the OOD region. This highlights the value of *data-centric* strategies that emphasize the generative factors in isolation, consistent with the finding that dataset augmentation with stripes also improves compositional generalization and data-efficiency scaling for 2D bump generation in diffusion models in Ref. (Liang et al., 2024b). While the prior work concluded that stripe training generated disentangled latent representations, the present work shows that supplying or learning disentangled latents is not sufficient for compositional generalization, and that stripe training even with disentangled latents helps form pixel-space latents representations that are necessary for OOD compositional generalization.

**Summary.** The commonality between the two disparate approaches that enable OOD generalization is that they both encourage the network to form *disentangled representations that are encoded in the space of the 2D pixel outputs*. By contrast, even directly supplying the network with factorized representations does not work if those representations are abstract and in a different latent space than the 2D image space. Factorization in a latent space may not survive subsequent dimension expansion by a downstream decoder. By ensuring that the network learns to separate each underlying factor (e.g. $x$ and $y$) at the pixel level, we preserve the necessary compositional structure for strong extrapolation to novel conditions.

## 5. Discussion & Conclusion

Our results highlight two overarching principles:

1. **Disentangled latents are not enough.** Factorizing a single latent layer (e.g. the bottleneck) into a disentangled representation does not guarantee compositional structure in deeper layers; manifold warping can re-entangle factors, leading to failure in OOD regions.
2. **Rendering disentangled latents into the same representational space as the outputs is key.** Through a specific architecture with low-rank regularization or

domain-centric data curation (training with independent factors), the network generates pixel-level embeddings of the individual $x$, $y$ latents to achieve strong compositional generalization.

**Limitations & Future Directions.** While our investigation provides a clean, mechanistic insight into why disentangled representations often fail on a synthetic 2D bump task, its scope is inherently limited by the toy nature of the setting. Extending these insights to large-scale models (e.g., diffusion models, autoregressive architectures, Transformers) and high-dimensional data domains (e.g., vision, language) is an important next step. Modern architectures like vision transformers or large language models may already embed implicit factorization biases—through attention mechanisms or prompt-based modularity—but rigorously assessing whether these biases genuinely support compositional generalization remains an open challenge.

We also acknowledge that translating insights from our synthetic study to more complex, real-world datasets is nontrivial but essential. As highlighted by Montero et al. (2022), interactive compositionality is unlikely to be addressed by a one-size-fits-all solution. Accordingly, we do not claim our approach is universally applicable, but rather that it reveals two promising and generalizable directions: (1) training modular, output-level embedding filters dedicated to each disentangled input dimension, and (2) leveraging dataset augmentation strategies that isolate factors of variation. Our simplified setting was deliberately chosen to cleanly expose the underlying failure modes, such as the model memorizing ID configurations for OOD generalization, without confounding interactions.

In future work, we plan to explore how these principles scale to standard disentanglement datasets and more naturalistic settings, explicitly investigating whether modular embedding and low-rank factorization constraints retain their utility. Finally, we emphasize the need for new metrics and frameworks to characterize *partial* and *hierarchical* compositional structures, which are far more representative of real-world cognition. We provide a preliminary attempt at capturing such structures in Appendix A. Developing principled tools to measure subtler forms of compositionality may better guide both architectural and algorithmic choices in the pursuit of robust generalization in complex domains.

Ultimately, bridging the gap between *disentangled representations* and *compositional generalization* requires a holistic view of how a network's *entire* forward pass maintains or destroys factorization. We hope this work provides a step toward that goal, clarifying both the pitfalls of standard bottleneck-based approaches and the promise of structural or regularized solutions.

## Acknowledgements

The authors would like to thank Alan (Junzhe) Zhou, Ziming Liu, Federico Claudi, Megan Tjandrasuwita, Mitchell Ostrow, Sarthak Chandra, Ling Liang Dong, Hao Zheng, Tomaso Poggio, Cheng Tang for helpful discussions and feedback at various stages of this work.

## Impact Statement

Our work develops theoretical and empirical insights into why disentangled or factorized inputs do not necessarily yield compositional generalization, and proposes practical strategies (data augmentation, low-rank regularization) to preserve factorization within the network. By illuminating how and why factorization can break down—or be maintained—this research can guide future model designs that are more transparent, data-efficient, and capable of robust extrapolation beyond the training distribution.

From an ethical or societal standpoint, we do not foresee immediate risks arising directly from these factorization techniques: the methods and analyses we present are primarily at the conceptual and architectural level. However, we note that improvements in compositional generalization can enable models to be more adaptable, including in sensitive real-world domains such as medical imaging or autonomous systems. While better extrapolation is desirable, it also entails that models could behave in unintended ways under distribution shifts if they mix factors in novel, untested combinations. Ultimately, we believe our framework, by clarifying the conditions needed for truly factorized representations, will enhance the interpretability and reliability of neural networks, a net positive for the broader machine learning community.

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

# A. A Framework for Compositionality, Factorization, and Modularity

Existing literature is divided on whether disentangled representation learning truly fosters compositional generalization and data efficiency. Despite extensive empirical studies and numerous proposed metrics, there is still no unified framework that clearly defines factorization, disentanglement, and compositionality or their interrelationships. A key challenge lies in the lack of a systematic "language" to characterize the various types and degrees of these concepts. Furthermore, real-world datasets often exhibit multiple, overlapping forms of compositionality, each with its own failure modes that can obscure one another. In response, we aim to develop a more general framework that makes these distinctions explicit.

This section aims to establish a framework for unifying the concepts of compositionality, factorization, disentanglement, and modularity. Specifically, we introduce two core components of our framework. First, we define the *compositional complexity* of a target function or distribution via the minimal number of rank-1 product terms needed to represent it. Then, we formalize *factorization* (disentanglement) for the intermediate activations of a neural network, allowing for arbitrary invertible transformations.

## A.1. Definition of compositionality and compositional complexity

Here we take on a function learning perspective of defining factorization and compositionality. Specifically, for learning a compositional solution, we expect network to first learn how to "break down" (factorize) component factors and then flexibly "re-combine" them for compositional generation. For a neural network learning a generic $N$-way composite target function $f : \mathcal{X}_1 \times \cdots \times \mathcal{X}_N \to \mathbb{R}^M$ of $N$ features $\{x_1, x_2, \ldots, x_N\}$, we can write the tensor decomposition of this function as

$$f(x_1, \ldots, x_N) = \sum_{r=1}^{R} \lambda_r \, \phi_r^{(x_1)}(x_1) \cdots \phi_r^{(x_N)}(x_N), \tag{13}$$

where $\phi_r^{(x_i)}(x_i) : \mathcal{X}_i \to \mathbb{R}^M$ and $\lambda_i \in \mathbb{R}$ are lumped factors.

Note that this decomposition is not unique, meaning that a given function $f$ might permit multiple degenerate decomposition with the same rank or different ranks. This means that the network is not guaranteed to learn a compositional decomposition that aligns with the canonical axes of variation, as required by some of the disentanglement metrics in existing literature.

**Probability Distributions as a Sum of Product Factors.** In some settings, rather than learning a deterministic function $f : \mathcal{X}_1 \times \cdots \times \mathcal{X}_N \to \mathbb{R}^M$, we aim to model a *joint probability distribution* $p(x_1, \ldots, x_N)$. An analogous "sum-of-products" factorization can be written as:

$$p(x_1, \ldots, x_N) \;=\; \sum_{r=1}^{R} \lambda_r \, p_1^{(r)}(x_1) \, \cdots \, p_N^{(r)}(x_N), \quad \text{with} \quad \sum_{r=1}^{R} \lambda_r \;=\; 1, \quad \lambda_r \geq 0. \tag{14}$$

Here, each $p_i^{(r)}(x_i)$ denotes a univariate (or lower-dimensional) distribution on $\mathcal{X}_i$, and $\lambda_r$ are nonnegative mixture weights. In discrete domains, $p_i^{(r)}$ can be categorical distributions; in continuous domains, they can be densities (e.g. Gaussian components or neural-network–based PDFs). Intuitively, we are representing $p$ as a *mixture* of "fully factorized" product distributions, each contributing a rank-1 tensor $p_1^{(r)} \times \cdots \times p_N^{(r)}$ in the joint space.

**Defining Compositional Complexity.** There are some compositional functions or probability distributions that are inherently more difficult to learn than others. Here we formalize the "compositional complexity" of a target function or distribution via the minimal number of rank-1 (factorized) terms required in a sum-of-product decomposition. Let $N$ be the number of factors (or dimensions), and let $\{\phi_r^{(x_i)}\}$ or $\{p_i^{(r)}\}$ denote per-factor functions or distributions.

**Case 1: Deterministic Function.** For $f : \mathcal{X}_1 \times \cdots \times \mathcal{X}_N \to \mathbb{R}^M$, assume $M = 1$ for simplicity, or treat each output dimension independently. A sum-of-product (CP) representation is

$$f(x_1, \ldots, x_N) \;=\; \sum_{r=1}^{R} \prod_{i=1}^{N} \phi_r^{(x_i)}(x_i), \quad \text{(up to a scalar factor } \lambda_r \text{ if desired).} \tag{15}$$

Define the *exact rank* of $f$ as:

$$R^*(f) \;=\; \min\left\{ R \;\middle|\; f(x_1, \ldots, x_N) \;=\; \sum_{r=1}^{R} \prod_{i=1}^{N} \phi_r^{(x_i)}(x_i) \right\}. \tag{16}$$

If $f$ cannot be exactly decomposed with finite $R$, we consider an $\varepsilon$-approximation in some norm or error measure $\|\cdot\|$:

$$R_\varepsilon^*(f) \;=\; \min\left\{ R \;\middle|\; \left\| f - \sum_{r=1}^{R}\prod_{i=1}^{N}\phi_r^{(x_i)}(x_i) \right\| \;\leq\; \varepsilon \right\}. \tag{17}$$

A smaller $R^*(f)$ (or $R_\varepsilon^*(f)$) indicates $f$ is more "compositional" or factorized.

**Case 2: Probability Distribution.**  For a joint probability $p(x_1,\ldots,x_N)$, a sum-of-products corresponds to a finite mixture of product distributions:

$$p(x_1,\ldots,x_N) \;=\; \sum_{r=1}^{R}\lambda_r\prod_{i=1}^{N}p_i^{(r)}(x_i), \quad \sum_{r=1}^{R}\lambda_r = 1, \;\; \lambda_r \geq 0. \tag{18}$$

Its *exact rank* is:

$$R^*(p) \;=\; \min\left\{ R \;\middle|\; p = \sum_{r=1}^{R}\lambda_r\prod_{i=1}^{N}p_i^{(r)}(x_i) \right\}. \tag{19}$$

If no finite $R$ yields an exact factorization, we allow an $\varepsilon$-approximation w.r.t. a suitable divergence $\mathrm{D}(\cdot,\cdot)$ (e.g. KL divergence or total variation distance):

$$R_\varepsilon^*(p) \;=\; \min\left\{ R \;\middle|\; \mathrm{D}\!\left(p \;\middle\|\; \sum_{r=1}^{R}\lambda_r\prod_{i=1}^{N}p_i^{(r)}\right) \;\leq\; \varepsilon \right\}. \tag{20}$$

Again, smaller ranks correspond to "low compositional complexity" (fewer factorized terms), whereas large $R$ indicates more intricate cross-factor entanglements.

### A.2. Definition of Factorization/Disentanglement of Neural Networks

Having introduced the notion of *intrinsic compositionality* for a target function or distribution, we now shift to defining *factorization* (a.k.a. disentanglement) in the context of a neural network's activation space. This concept applies to an intermediate layer or the entire network.

**Defining factorization vs. disentanglement.**  Despite the frequent usage of these terms, there is no universal consensus on their precise definitions in the literature. Some works treat them as strictly synonymous, while others reserve "factorization" for representations whose coordinates directly correspond to physically or semantically independent factors (e.g. each latent dimension controlling one ground-truth variable). By contrast, "disentanglement" can refer to a broader class of methods or criteria (e.g. axis-aligned independence, mutual information scores, or partial correlations). In this paper, we do not distinguish sharply between these nuances. Wherever we refer to "disentangled" or "factorized" representations, we mean that a model's latent (or intermediate) dimensions reflect independently varying components of the data. The specific implementation details—whether formalized via a $\beta$-VAE objective, mutual-information measures, or other constraints—are secondary to the overall idea that each latent dimension ideally captures a different underlying factor. Hence, we use the two terms *interchangeably* throughout the manuscript.

**Activation Matrix Setup.**  Let

$$\alpha: \; \mathcal{X}_1 \times \cdots \times \mathcal{X}_N \;\to\; \mathbb{R}^D \tag{21}$$

represent the activations of a given network layer. For an input $(x_1,\ldots,x_N)$, the output $\alpha(x_1,\ldots,x_N) \in \mathbb{R}^D$. Over $N_{\mathrm{ID}}$ training samples $\{(x_{1,i},\ldots,x_{N,i})\}_{i=1}^{N_{\mathrm{ID}}}$, we can assemble an activation matrix

$$\alpha \;\in\; \mathbb{R}^{N_{\mathrm{ID}} \times D}, \quad \text{where row } i \text{ is } \alpha(x_{1,i},\ldots,x_{N,i}). \tag{22}$$

Optionally, we may project $\alpha$ to a lower dimension $K \leq D$ via $W \in \mathbb{R}^{K \times D}$ (e.g. PCA), yielding

$$\alpha_{\text{feat}}(x_1,\ldots,x_N) \;=\; \alpha(x_1,\ldots,x_N)\, W^\top \;\in\; \mathbb{R}^K. \tag{23}$$

**General Definition of Factorization.** A purely linear test might require an invertible matrix $U \in \mathbb{R}^{K \times K}$ such that

$$\alpha_{\text{feat}}(x_1, \ldots, x_N) U^\top \; = \; \big[\beta_1(x_1), \ldots, \beta_N(x_N)\big], \tag{24}$$

where each block of coordinates depends *only* on a single factor $x_j$. However, factorization may emerge only under a *nonlinear* reparametrization. Hence, we relax the requirement to any invertible map

$$T : \; \mathbb{R}^D \; \to \; \mathbb{R}^D \quad \text{(e.g. a diffeomorphism or normalizing flow).} \tag{25}$$

**Definition (General Factorization).** We say $\alpha : \mathcal{X}_1 \times \cdots \times \mathcal{X}_N \to \mathbb{R}^D$ is *nonlinearly factorized* w.r.t. $(x_1, \ldots, x_N)$ if there exist:

1. An invertible transformation $T : \mathbb{R}^D \to \mathbb{R}^D$,
2. A partition of $\{1, \ldots, D\}$ into $N$ disjoint subsets $I_1 \cup \cdots \cup I_N$,

such that defining

$$\beta(x_1, \ldots, x_N) \; = \; T\big(\alpha(x_1, \ldots, x_N)\big) \; \in \; \mathbb{R}^D, \tag{26}$$

we have, for every coordinate $i \in I_j$,

$$\beta_i(x_1, \ldots, x_N) \; = \; \beta_i(x_j) \quad \text{depends } only \text{ on } x_j. \tag{27}$$

Concretely, in these transformed coordinates $(\beta_1, \ldots, \beta_N)$, each subset of coordinates implements the feature(s) of a single factor $x_j$, with *no cross-dependence* among different factors.

**Modularity via Factorization.** A representation is *functionally modular* if, under some invertible transform $T$, we can rearrange the coordinates into sub-blocks that each depend on exactly one factor $x_j$. This directly matches the factorization criterion above. In contrast, a *structurally modular* network places each factor's neurons in physically separate submodules (e.g. block-diagonal connectivity), making the separation visible *without* any transform. Hence, functional modularity (factorization) is more general and only requires a suitable reparametrization in $\mathbb{R}^D$, whereas structural modularity is a stricter design property at the raw parameter level.

### A.3. Partial Factorization, Entanglement, and Reversibility

Sections A.2 gave a definition of factorization vs. entanglement: if *any* invertible map $T$ can reorder the coordinates into blocks that depend purely on $x$ or purely on $y$, we call the representation factorized; otherwise, it is entangled. In practice, however, real-world data and deep networks often produce *partial* factorization, where some features mix $x$ and $y$ only mildly or separate them only in certain regions. Further, certain layers introduce *irreversible* steps (e.g. pooling, saturations), which can destroy factor structure. We elaborate on these points below.

**Partial Factorization.** A representation need not split perfectly into two blocks ($I_x$ and $I_y$) to convey some compositional benefits. For example, consider a multi-factor domain $(x, y, z)$ in which the representation merges two of the factors $(x, y)$ but leaves $z$ separate. One might say the network has *partially factorized* the input. More formally, partial factorization arises if *some* invertible map $T$ can partition the $D$ coordinates into sub-blocks, each depending on a subset (not necessarily size 1) of the factors. This differs from pure factorization in that each block may mix more than one factor, yet remain independent of other blocks. Concretely:

$$T\big(\alpha(x, y, z)\big) \; = \; \big[\beta_{xy}(x, y), \; \beta_z(z)\big], \quad \text{with minimal cross-dependence among blocks.} \tag{28}$$

This partial structure still yields some compositional benefits (e.g. reusing the $(x, y)$ sub-block in new contexts of $z$). But it may not qualify as fully "disentangled" in the strict sense of Section A.2. Indeed, such partial factorization commonly appears if $x$ and $y$ are correlated in training data or if the network's architecture merges them early.

**Entanglement via Irreversible Mixing.** Even if a representation starts factorized in an early layer, the network may subsequently *entangle* factors by applying irreversible operations such as:

- **Pooling or Striding:** Combining multiple spatial positions (or multiple channels) into one, discarding the ability to invert them individually.

- **Nonlinear Saturation:** A ReLU or sigmoid can "clip" signals, so the original amplitude differences from $x$ vs. $y$ can no longer be fully recovered.

- **Dimension Reduction:** Fully connected layers that collapse from a higher dimension $D$ to a smaller one $D' < D$ can forcibly merge features.

Because our definition of factorization requires an *invertible* map $T$ to unscramble the coordinates, any irreversible merge that loses information effectively prevents us from re-separating $x$ and $y$. Thus, irreversible mixing is a key mechanism by which networks destroy factorization.

**Reversible Mixing Need Not Break Factorization.** By contrast, a *reversible* or *bijective* layer can combine factors in intermediate channels without permanently entangling them, so long as there exists an inverse transform. For instance, a normalizing-flow block may shuffle or mix $(x, y)$ channels in a fully invertible manner; the overall system can still preserve factorization if the subsequent training does not "pinch" those degrees of freedom. In principle, any invertible map that merges coordinates can be "undone" later, so the final representation might remain factorized.

### A.4. A Kernel-Based Perspective on Factorization

To operationalize the above definitions of factorization/disentanglement concretely, we here propose a kernel-based perspective and define a *factorization metric* that assess the factorization of the kernel learned by the neural network. The goal here is to characterize such a kernel, given its potentially factorized form, to understand the different regimes in which it compositionally generalize (or not).

We now consider an *N-partite* domain $\mathcal{X}_1 \times \cdots \times \mathcal{X}_N$ and a positive-semidefinite (PSD) kernel

$$K : \left( \mathcal{X}_1 \times \cdots \times \mathcal{X}_N \right) \times \left( \mathcal{X}_1 \times \cdots \times \mathcal{X}_N \right) \to \mathbb{R}. \tag{29}$$

Under suitable conditions (e.g. an analogue of Mercer's theorem), $K$ often admits a *sum-of-product* (canonical polyadic) expansion:

$$K\left( (x_1, \ldots, x_N), (x'_1, \ldots, x'_N) \right) = \sum_{r=1}^{R^*} \lambda_r \prod_{j=1}^{N} k_j^{(r)}(x_j, x'_j), \tag{30}$$

where each term $\lambda_r \prod_{j=1}^{N} k_j^{(r)}(x_j, x'_j)$ is *rank-1* across the $N$ factors, and $R^*$ may be finite or infinite.

**Interpretation.** If $R^* = 1$ in (30), we say $K$ is a *purely factorized* (rank-1) kernel, effectively separating each factor $x_j$. Larger $R^* > 1$ signals *partial* or *full* entanglement across these factors in $K$.

**Rank-Based Measure of Factorization.** We define the *CP rank* of $K$ by

$$\mathrm{rank}(K) = \min\{ R^* \mid (30) \text{ holds exactly} \}, \tag{31}$$

in analogy to the Schmidt rank in bipartite systems. A small rank implies that $K$ is closer to a "fully separable" structure across $(x_1, \ldots, x_N)$, while a large rank means more cross-term mixing.

### A.5. Polynomial Expansions and a Generalized Factorization Metric

Suppose $K$ admits a polynomial-like expansion enumerating single-factor, pairwise, and higher-order cross terms:

$$K = \sum_{\ell} \theta_{\ell} K_{\ell}, \tag{32}$$

where each $K_{\ell}$ might be an $n$-factor product kernel on a subset of coordinates, and $\theta_{\ell}$ is its coefficient. We assign each $\theta_{\ell}$ a "weight" $w(\theta_{\ell})$, for instance $|\theta_{\ell}|$ or $\theta_{\ell}^2$. A *generalized factorization metric* then measures the fraction of $K$'s "weight" contributed by a chosen subset of "lower-order" terms. Formally, if $\mathcal{S}$ denotes the subset of indices corresponding to single-factor (or low-order) terms, we define

$$\mathrm{PF\_Coeff}^{(\mathcal{S})}(K) = \frac{\sum_{\ell \in \mathcal{S}} w(\theta_{\ell})}{\sum_{\ell} w(\theta_{\ell})}. \tag{33}$$

By varying $\mathcal{S}$ (e.g. only single-factor vs. single-plus-pairwise), we obtain a family of "explained fraction" values—analogous to cumulative explained variance in PCA. A higher fraction indicates that $K$ is dominantly factorized by low-order expansions (less entangled), whereas large higher-order terms imply more cross-variable mixing.

**Operational Regimes of NNs in light of this decomposition.** Based on the defined rank-based measure of factorization, we can imagine two operational regime limits of the neural networks, namely

- *Pure memorization regime*: in which case the network resorts to tabular learning (memorization), i.e. learning a single $N$-factor kernel term

- *Pure factorization regime*: in which case the network learns a completely compositional kernel (assuming each of the $N$ factors are independent).

These are obviously theoretical limits, and in reality, the network probably operates in a regime that interpolates between the two, i.e. learning partially/imperfectly factorized solutions.

**How to estimate this metric in practice.** In real-world scenarios, we rarely have direct access to the continuous kernel $K : (\mathcal{X}_1 \times \cdots \times \mathcal{X}_N)^2 \to \mathbb{R}$. Instead, we can discretize $K$ by selecting a finite set of data points $\{(x_{1,i}, \ldots, x_{N,i})\}_{i=1}^M$ and forming the $M \times M$ *Gram matrix*:

$$\mathbf{K}_{(i,j)} \;=\; K\Big(\big(x_{1,i}, \ldots, x_{N,i}\big), \big(x_{1,j}, \ldots, x_{N,j}\big)\Big). \tag{34}$$

We can interpret this matrix as a discrete approximation to $K$. To apply our *generalized factorization metric* (§A.5), we then:

1. **Approximate expansions.** Attempt to factor (or partially factor) $\mathbf{K}$ into a sum-of-product form among the $N$ factors, e.g. through low-rank tensor methods or suitable polynomial expansions on the sampled coordinates.

2. **Coefficient extraction.** If we assume a polynomial-like expansion, $\mathbf{K} \approx \sum_\ell \theta_\ell \mathbf{K}_\ell$, we obtain finite-dimensional coefficients $\{\theta_\ell\}$ by fitting or projecting onto a chosen basis.

3. **Compute the factorization ratio.** Restrict attention to the "low-order" terms (e.g. single-factor or pairwise) to form a subset $\mathcal{S}$. Define

$$\mathrm{PF\_Coeff}^{(\mathcal{S})}(\mathbf{K}) \;=\; \frac{\sum_{\ell \in \mathcal{S}} w(\theta_\ell)}{\sum_\ell w(\theta_\ell)}, \tag{35}$$

just as in the continuous setting, but now $\ell$ indexes a finite set of basis or fitted components.

This procedure yields a practical, data-driven measure of how "factorized" the *empirical kernel* is on the sampled domain. Notably, if the sampling is sparse or unrepresentative, the resulting $\mathbf{K}$ may fail to capture subtle cross-factor structure. In that sense, one can view $\mathrm{PF\_Coeff}^{(\mathcal{S})}(\mathbf{K})$ as an *approximate* or *lower-dimensional* proxy for $\mathrm{PF\_Coeff}^{(\mathcal{S})}(K)$ in the continuous limit.

## B. Metric Tensor and Factorization Metric for $N$-Dimensional Inputs

In this appendix, we outline a generic procedure to quantify the local geometry of a neural representation $\mathbf{f}$ defined on an $N$-dimensional input space, with the metric tensor $g$ from Ref. (Zavatone-Veth et al., 2023), and then specialize to the 2D case with an explicit factorization metric that we propose based on $g$.

### B.1. General $N$-Dimensional Definition
**Representation and Jacobian.** Let

$$\mathbf{f} : \mathbb{R}^N \to \mathbb{R}^D, \tag{36}$$

where $\mathbf{f}(x_1, \ldots, x_N) = (f_1, \ldots, f_D)$ for each input vector $\mathbf{x} = (x_1, \ldots, x_N)$. We define the Jacobian $J$ of $\mathbf{f}$ at a point $\mathbf{x}$ by

$$J(\mathbf{x}) \;=\; \begin{bmatrix} \partial f_1/\partial x_1 & \cdots & \partial f_1/\partial x_N \\ \vdots & \ddots & \vdots \\ \partial f_D/\partial x_1 & \cdots & \partial f_D/\partial x_N \end{bmatrix} \;\in\; \mathbb{R}^{D \times N}. \tag{37}$$

**Metric Tensor.** The induced *metric tensor* $g(\mathbf{x})$ is then an $N \times N$ matrix,

$$g(\mathbf{x}) = J(\mathbf{x})^\top J(\mathbf{x}), \tag{38}$$

where $g_{\mu\nu}(\mathbf{x}) = \sum_{i=1}^{D}(\partial f_i/\partial x_\mu)(\partial f_i/\partial x_\nu)$. Geometrically, $g$ captures how infinitesimal distances in input space map to distances in the representation space $\mathbb{R}^D$. By computing $g$, we gain insights into how the native space becomes wraped in the embedding space (Fig. 5).

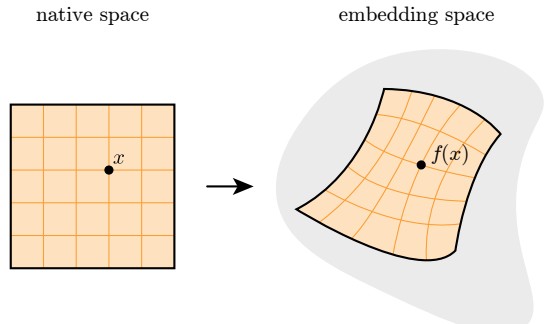

*Figure 5.* The action of the neural net can be imagined as wrapping of the (originally flat) native space (left) into the embedding space (right).

**Volume Element.** The local volume element is given by

$$dv(\mathbf{x}) = \sqrt{\det(g(\mathbf{x}))} \, dx_1 \ldots dx_N. \tag{39}$$

This indicates how the network "stretches" or "distorts" volumes near each $\mathbf{x}$.

**Factorization.** While there is no single universal scalar for "factorization" in the $N$-dimensional case, one can investigate off-diagonal elements of $g_{\mu\nu}$ (or sub-blocks of the Jacobian) to see how strongly certain input dimensions $(x_\mu)$ mix with others $(x_\nu)$. For instance, if $\partial\mathbf{f}/\partial x_\mu$ is approximately orthogonal to $\partial\mathbf{f}/\partial x_\nu$ for all $\mu \neq \nu$, the representation preserves independence among coordinates.

### B.2. Specialization to the 2D Case

When $N = 2$, let the input be $(x, y)$. Then the metric $g$ is a $2 \times 2$ matrix:

$$g = \begin{bmatrix} \langle \frac{\partial \mathbf{f}}{\partial x}, \frac{\partial \mathbf{f}}{\partial x} \rangle & \langle \frac{\partial \mathbf{f}}{\partial x}, \frac{\partial \mathbf{f}}{\partial y} \rangle \\ \langle \frac{\partial \mathbf{f}}{\partial y}, \frac{\partial \mathbf{f}}{\partial x} \rangle & \langle \frac{\partial \mathbf{f}}{\partial y}, \frac{\partial \mathbf{f}}{\partial y} \rangle \end{bmatrix}. \tag{40}$$

Its volume element is

$$dv(x, y) = \sqrt{\det(g(x, y))} \, dx \, dy. \tag{41}$$

**A Simple Factorization Metric in 2D.** If one desires a single scalar capturing how "uncoupled" $x$ and $y$ remain, we can define:

$$\text{Factorization}(g(x, y)) = 1 - \frac{|g_{xy}(x, y)|}{|g_{xx}(x, y)| + |g_{yy}(x, y)|}, \tag{42}$$

where $g_{xx} \equiv g_{0,0}$, $g_{yy} \equiv g_{1,1}$, and $g_{xy} \equiv g_{0,1} = g_{1,0}$. In words:

- $g_{xx}$ and $g_{yy}$ measure how much the representation changes when moving purely in the $x$- or $y$-direction.

- $g_{xy}$ captures cross-direction entanglement.

- The closer $\text{Factorization}(g)$ is to 1, the more orthogonal (uncoupled) these directions are in $\mathbb{R}^D$.

By averaging this score over a grid of $(x, y)$ points, one obtains a global measure of whether the representation remains factorized or becomes entangled.

Without Regularization          With Regularization

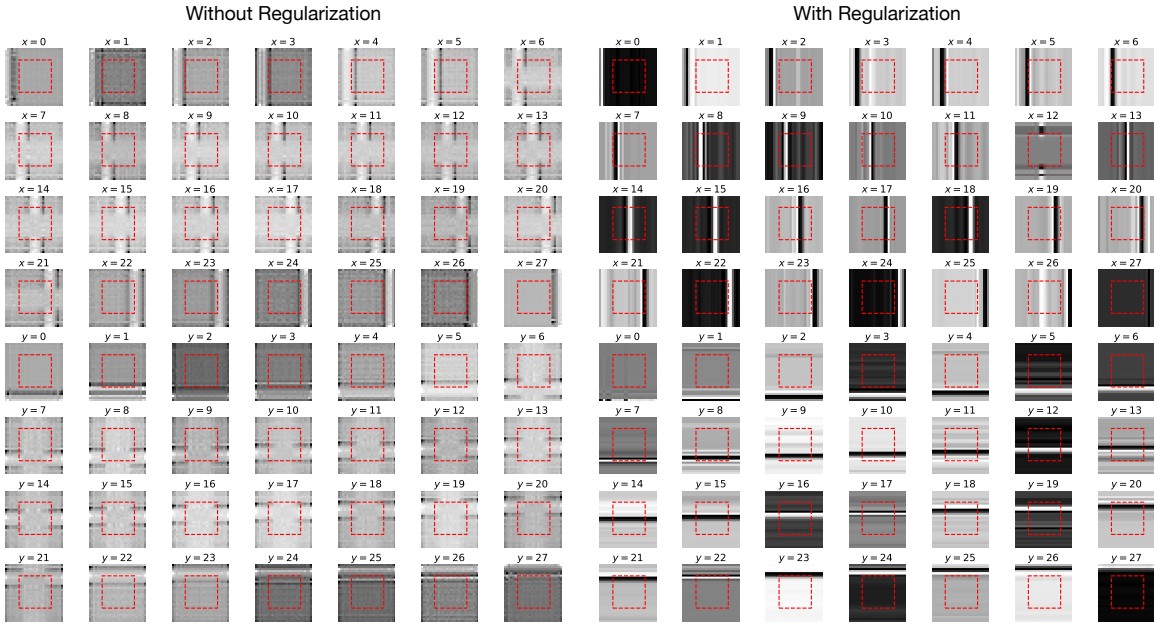

*Figure 6.* **Visualization of $x$ and $y$ embedding matrices without and with regularization.**

## C. Low-rank Regularization for Compositional Generalization

In our generative approach, each input factor (e.g. the one-hot code for coordinate $x$ or $y$) is mapped to a two-dimensional embedding matrix. Concretely, suppose we have a factor $\gamma$ (which can be $x$ or $y$), and we define a matrix

$$W_{i,j}(\gamma) \ \in \ \mathbb{R}^{d \times d}, \tag{43}$$

where the indices $(i, j)$ correspond to a row and column in this 2D embedding. The matrix $W(\gamma)$ is then passed, possibly along with other embeddings, into subsequent layers (e.g. a small convolutional network) to produce the final output image.

**Low-Rank Decomposition.** To encourage each embedding matrix $W_{i,j}$ to remain factorized, we represent it via an SVD-like decomposition:

$$W_{i,j} \ = \ \sum_{n=1}^{r} v_i^{(n)} \, \lambda^{(n)} \, u_j^{(n)}, \tag{44}$$

where $r$ is an upper bound on the rank, $\lambda^{(n)} \in \mathbb{R}$ are scalar coefficients, and $v_i^{(n)}, u_j^{(n)}$ are the row- and column-basis vectors for the $n$-th mode. We then regularize these parameters to encourage a low-rank and spatially consistent solution.

**Entropy and Variance Penalties.** First, we interpret the squared magnitudes $[\lambda^{(n)}]^2$ as a probability distribution across $n$,

$$\widetilde{\lambda}^{(n)} \ = \ \frac{[\lambda^{(n)}]^2}{\sum_{m=1}^{r}[\lambda^{(m)}]^2}, \tag{45}$$

and add an *entropy* penalty (DeMoss et al., 2024)

$$\mathcal{L}_{\text{ent}} \ = \ -\eta_1 \sum_{n=1}^{r} \widetilde{\lambda}^{(n)} \, \ln\!\big[\widetilde{\lambda}^{(n)}\big]. \tag{46}$$

Minimizing $\mathcal{L}_{\text{ent}}$ reduces the distribution's entropy, making only a few modes dominant (i.e. low rank).

Second, we penalize large row- or column-wise variance of the vectors $v^{(n)}, u^{(n)}$. Concretely, let $\text{Var}(v^{(n)})$ denote the sample variance of $v^{(n)}$ across indices $i$. Then we define

$$\mathcal{L}_{\text{var}} \ = \ \eta_2 \sum_{n=1}^{r} \Big[ \text{Var}\big(v^{(n)}\big) \ + \ \text{Var}\big(u^{(n)}\big) \Big]. \tag{47}$$

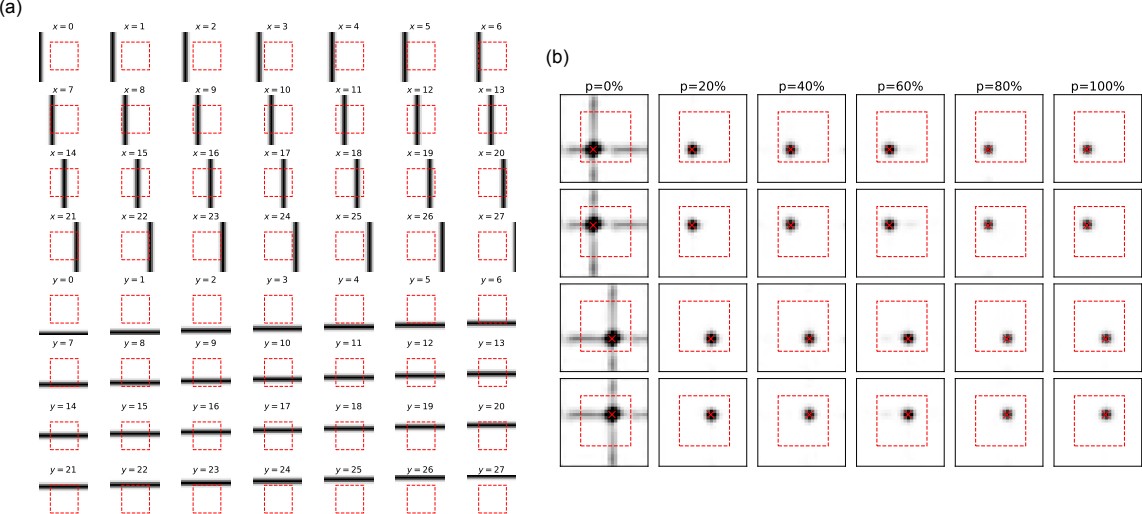

*Figure 7.* **Augmenting 2D Gaussian bump dataset with 1D Gaussian stripes enables compositional generalization.** (a) shows the 1D Gaussian stripe dataset for $N = 28$. (b) shows 4 sample generated OOD images for models trained on 1D Gaussian stripes + $p\%$ ID 2D Gaussian bumps subsampled.

Minimizing $\mathcal{L}_{\text{var}}$ discourages the model from "memorizing" individual entries $(i, j)$, thus promoting spatial invariance. Summing both these terms along with the usual reconstruction loss (e.g. MSE) yields

$$\mathcal{L}_{\text{total}} \;=\; \mathcal{L}_{\text{MSE}} \;+\; \mathcal{L}_{\text{ent}} \;+\; \mathcal{L}_{\text{var}}. \tag{48}$$

By applying (44)–(46) to the embedding matrices for both $x$- and $y$-factors, the network learns a more factorized, compositional representation that generalizes better to unseen coordinate combinations. Fig. 6 demonstrate the 2D embedding matrices corresponding to different values of $x$ and $y$ in the cases *without* and *with* the entropy + variance regularization. In the case of no regularization, the model fails to preserve the integrity of the signal across the OOD region. In contrast, with regularization, each embedding matrix exhibits more pronounced vertical or horizontal banding that aligns cleanly with the chosen $x$- or $y$-coordinate. This indicates the network has discovered a more factorized structure, making it easier to recombine $x$ and $y$ to generate a 2D Gaussian bump in unseen OOD configurations.

## D. Experimental Details

In this section, we describe the neural network architectures used for generating bump functions, including exact layer sizes, activation functions, optimization algorithms, hyperparameters, and training protocols. Code to reproduce the experiments can be found in this Github repository. We explored various input formats and model depths; below are detailed descriptions of representative CNN and MLP models using bump-encoding inputs:

| Property | CNN Architecture | MLP Architecture |
|---|---|---|
| Input Dimensions | 56 (reshaped to $56 \times 1 \times 1$) | 56 |
| Output Dimensions | $1 \times 28 \times 28$ | $1 \times 28 \times 28$ |
| Layers | 4 ConvTranspose2d layers (upsampling from $7 \times 7$ to $28 \times 28$) | 3 fully-connected linear layers |
| Hidden Layer Size | 64 channels per hidden layer | 256 units per hidden layer |
| Parameter Count | $\sim$315K | $\sim$272K |
| Activation Function | ReLU | ReLU |
| Optimizer | AdamW (Learning Rate: $1 \times 10^{-3}$) | AdamW (Learning Rate: $1 \times 10^{-3}$) |

*Table 1.* Comparison of CNN and MLP architectures

These details clearly communicate the architectural specifics and ensure reproducibility.

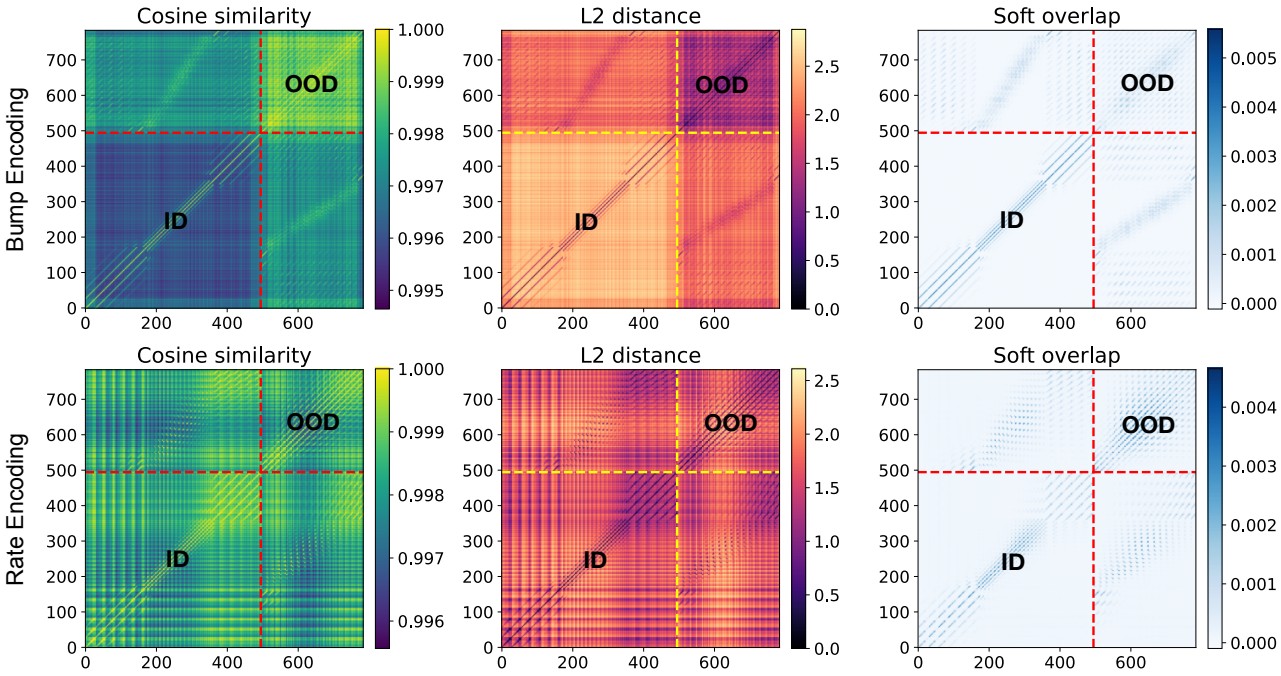

*Figure 8.* **Full similarity/distance/overlap matrices visualization across the entire ID-OOD data distribution, compared between networks with bump (top) and rate (bottom) input encodings.**

## E. Dataset Augmentation

One way of improving compositional generalization in our synthetic toy example is to *augment* the 2D Gaussian bump dataset with *1D Gaussian stripes*, thereby encouraging a more factorized representation in the pixel domain. Concretely, we generate *vertical* stripes by fixing a particular $x$-index and applying a 1D Gaussian along the $y$-axis, and *horizontal* stripes by fixing a $y$-index and applying a 1D Gaussian along the $x$-axis. Fig. 6(a) shows these stripes for $N = 28$, with each panel corresponding to a specific row or column coordinate. We then *combine* all stripes with a fraction $p\%$ of the original 2D bump dataset, so that examples of the 1D patterns are always present even when the 2D samples are subsampled.

This augmentation strategy allows the network to learn separate "axes" of variation (along $x$ and $y$) more explicitly. Empirically, as shown in Fig. 6(b), models trained on these augmented datasets exhibit improved extrapolation to out-of-distribution (OOD) regions, even with limited 2D bump data. The presence of 1D stripes encourages a form of compositional inductive bias, wherein the model learns to *additively* combine horizontal and vertical factors in the pixel domain. Consequently, we gain the insight that guiding models toward simpler, factorized building blocks can significantly enhance their ability to recombine learned components in novel settings, an essential capability for robust compositional generalization.

## F. Supplementary Experimental Results

### F.1. Full Kernel Characterization

In Sec. 3.3, we characterized the model's OOD generalization behavior from a kernel-based perspective. In Fig. 1(e), we showed the agreement between the similarity (overlap) matrix with the theorized binary factorized kernel, which showed that the model independently "processes" novel input combinations of $x$ and $y$. Here, we provide the full similarity/distance/overlap matrices across the entire generated dataset, ID and OOD. We note that Fig. 1(e) essentially shows a slice of what are shown in Fig. 8. We compare between generated image by models trained with bump-encoding and ramp-encoding inputs, and the matrices are constructed based on three metrics: 1) cosine similarity, 2) L2 distance, and 3) pixel overlap. We notice that the kernel matrices of bump-encoded model exhibit stronger "block" structure as compared

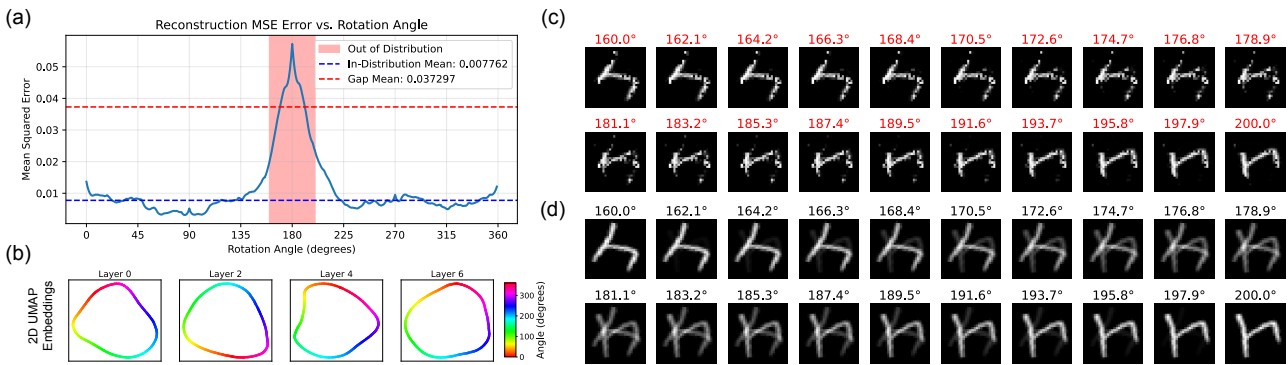

*Figure 9.* **MNIST digit "4" image rotation experiment with OOD range of** $160° − 200°$. (a) shows the reconstruction MSE as a function of rotation angle. (b) shows the 2D UMAP embeddings for representations at different layers of the network. (c) and (d) shows the generated rotated images in the OOD range vs. the pixel-wise interpolated images between $160°$ and $200°$.

to those of the ramp-encoded model. Here we note that the matrices have interesting diagonal structure from the non-finite width of the Gaussian bumps.

### F.2. MNIST Digit Image Rotation Experiments

In Sec. 3.3, we mentioned that the phenomenon of activation superposition corresponding to memorized ID data seem to be generically applicable beyond the compositional setting. In Fig. 9, we present a simple toy experiment on rotating a single MNIST image data of number "4." We generate a dataset of this rotated image "4" for various rotation angles $\theta \in [0, 360°]$ while leaving out an OOD range $\theta^{\text{OOD}} \in (160°, 200°)$. We then train a CNN on the ID data, with input $(\cos\theta, \sin\theta)$, and test if the model can interpolate/extrapolate to the $\theta^{\text{OOD}}$ range. Unsurprisingly, the model fails to generalize to the unseen rotation angle range, as shown by the MSE plot in Fig. 9(a). Nevertheless, the generated OOD images reveal that the model performs linear interpolation between the edge-most ID samples, namely the images rotated at $160°$ and $200°$. In Fig. 9(d), we show the linear interpolation between the ground truth rotated images at $160°$ and $200°$, which strongly resembles the generated OOD images shown in Fig. 9(c). Intriguingly, in Fig. 9(b) we show that the UMAP-embedded latent representations across the network layers remain smooth without significant deformation, unlike the case of 2D Gaussian bump generation presented in the main text. Nonetheless, the OOD outputs of the model as linear interpolation of ID images confirms that the model is indeed memorizing all ID images and superposing the corresponding activation patterns, which resonates with our findings for the 2D Gaussian bump generation experiment.

### F.3. Volume Metric Plots and Generated OOD Images

In Sec. 3.4, we characterized manifold warping using the Jacobian based volume metric, as well as factorization and linear probe metrics as a function of layer depth. Here we show the same metrics for networks of different depths, for both CNNs (Fig. 11 and Fig. 12) and MLPs (Fig. 13 and Fig. 14). We also show the generated images as a function of network depth for CNNs in Fig. 10. An interesting phenomenon here is that the MLP models seems to have the opposite "warping" effect in the OOD region as compared to the CNN models. Moreover, the MLPs seem to have a smoother but more drastic degradation of factorization as a function of layer depth. Nonetheless, we expect similar conclusions regarding disentanglement and factorization to hold for CNNs and MLPs alike.

### F.4. Neural Tuning Curves

In Sec. 4.2, we discussed the potential benefits of training models with augmented datasets that contain isolated factors of variation. Specifically, we showed that models trained with stripes+bumps are able to compositionally generalize in a data efficient manner. Mechanistically, the stripes provide the model a shortcut bias that leads the model to form additive composition in early layers of the network, which manifests as "stripe conjunctions," which become "bumps" in downstream layers. In Fig. 15, we visualize four layers of a CNN trained with stripes only (left) and with stripes+100 bumps (right). Each sub-panel shows a 2D activation map for a selected neuron, along with its mean activation when collapsed over $y$

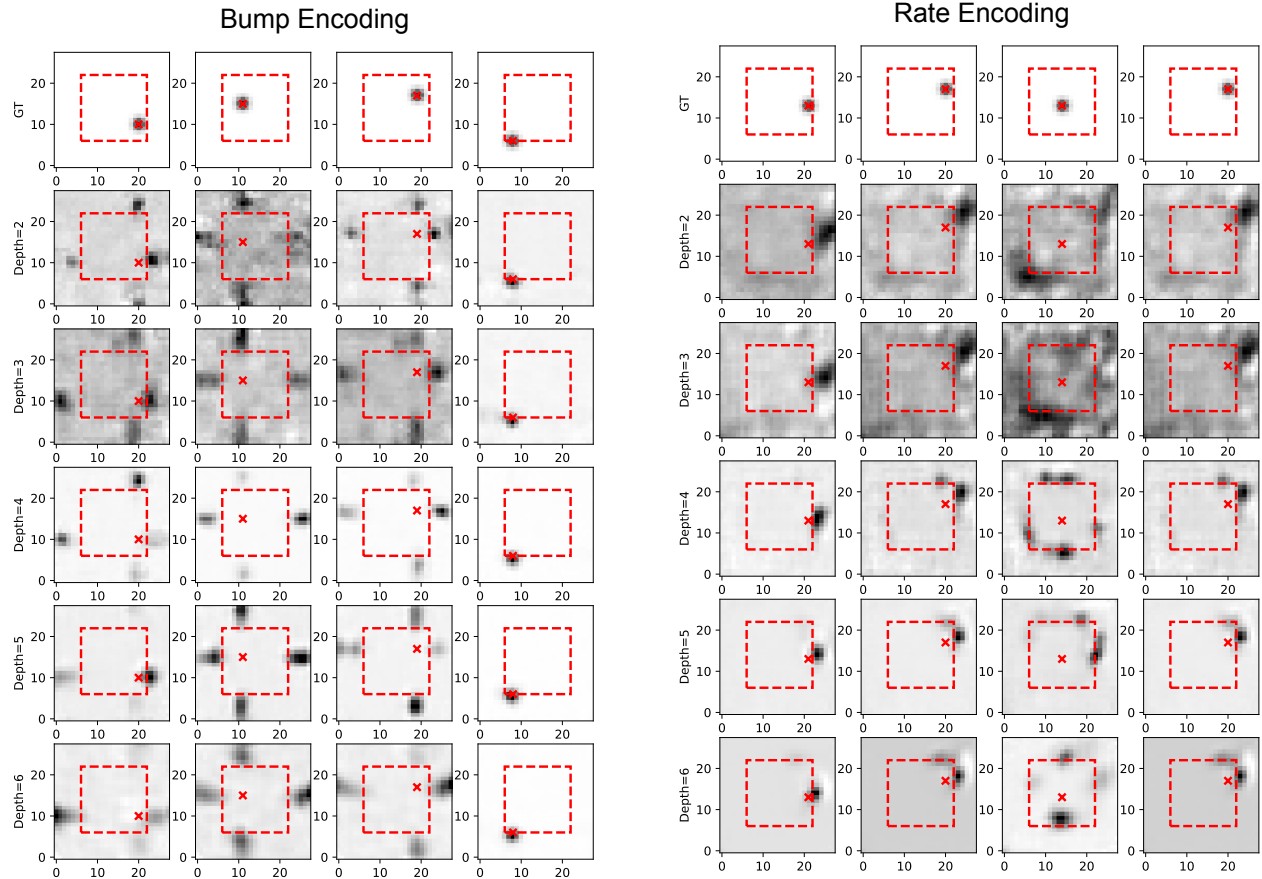

*Figure 10.* **Comparison of generated OOD images for different network depths between networks with bump-based (left) and ramp-based (right) input encodings.**

(middle plot) or $x$ (right plot). Across the layers, we observe that early-layer neurons primarily capture stripe-like features, with broader activations that vary more strongly along one dimension than the other. In deeper layers, these "stripes" begin to intersect more tightly, forming increasingly localized "bump"-like patterns. This indicates that the model is successfully learning to capture the complex compositional structures present in the augmented data, reflecting the additive-to-multiplicative transition in its internal representations. Furthermore, as bumps accumulate, the activations demonstrate a clear shift toward more focused and precise tuning, which supports the idea that the neural network is generalizing bump-like structures efficiently through additive composition early in the architecture and refining this understanding as the network deepens.

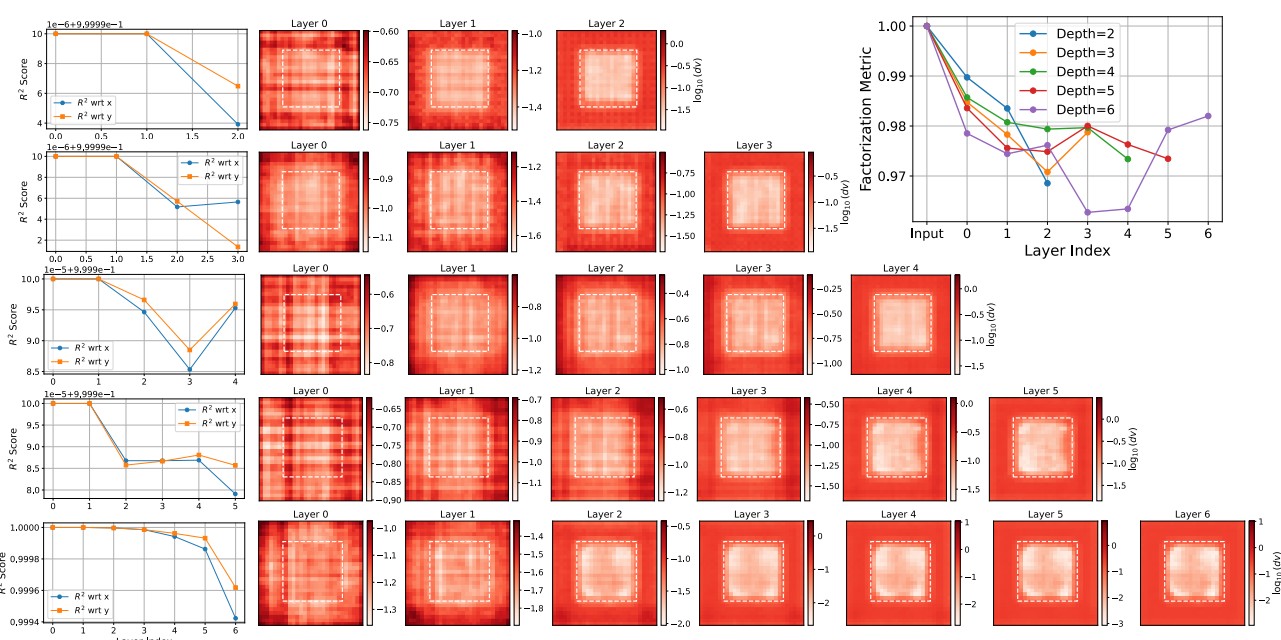

*Figure 11.* **Evolution of manifold density and linear probe as a function of layer depth for different CNN network depths (bump encoding).**

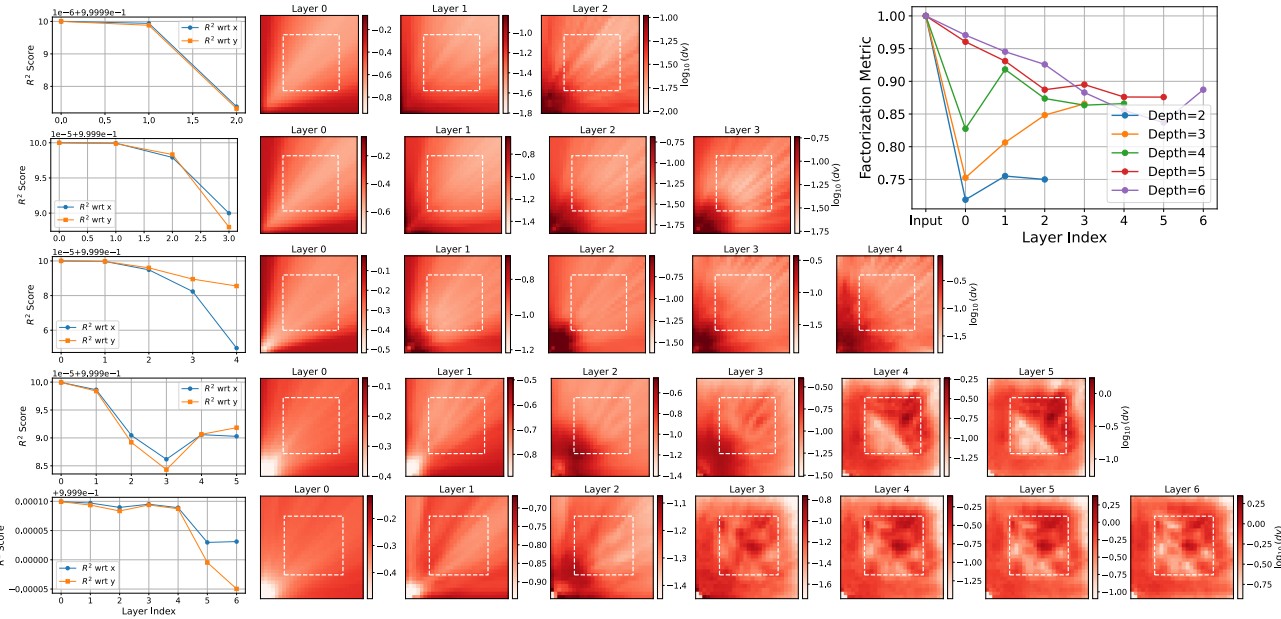

*Figure 12.* **Evolution of manifold density and linear probe as a function of layer depth for different CNN network depths (rate encoding).**

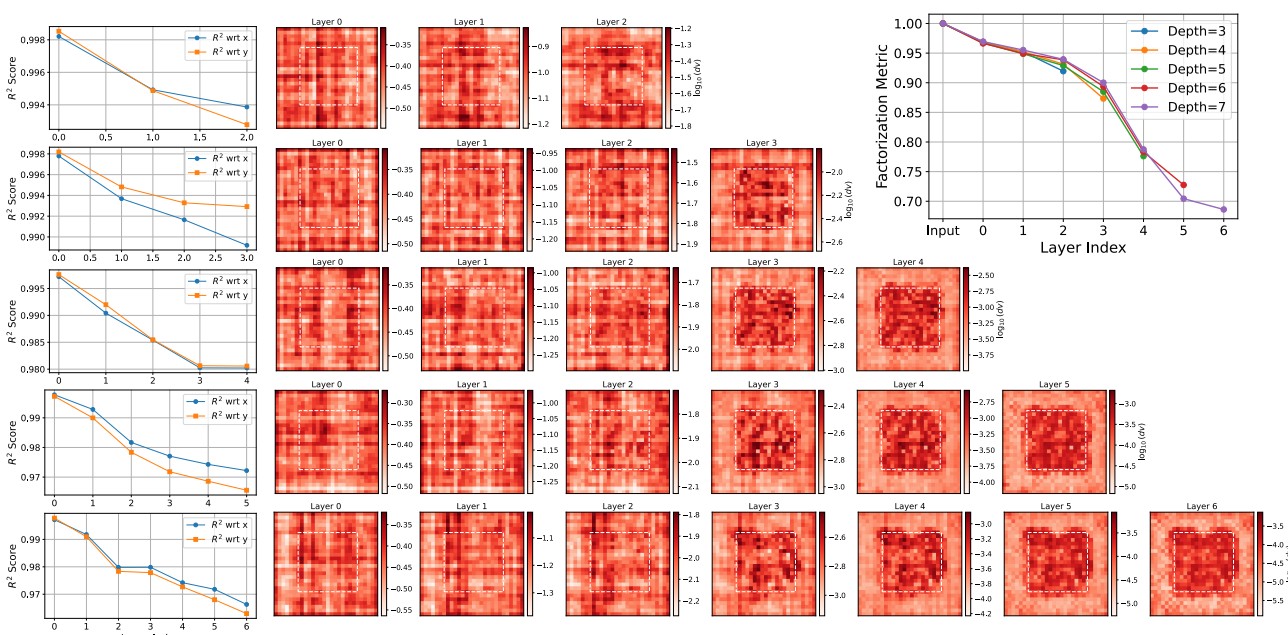

*Figure 13.* **Evolution of manifold density and linear probe as a function of layer depth for different MLP network depths (bump encoding).**

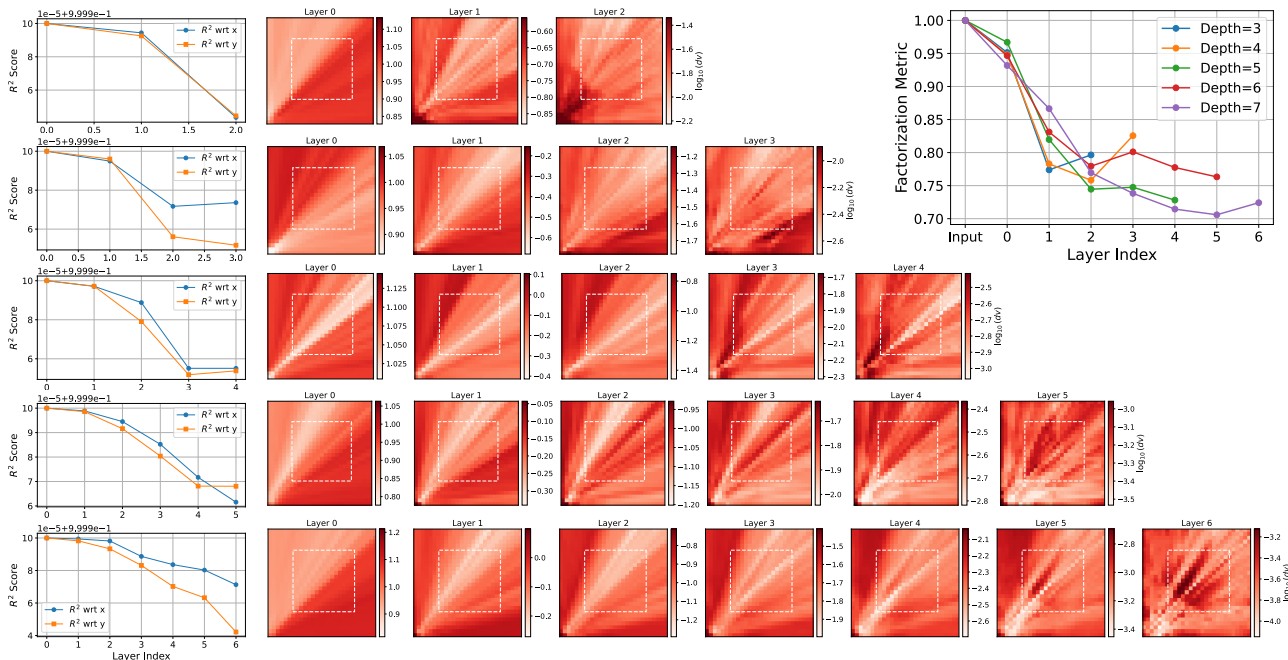

*Figure 14.* **Evolution of manifold density and linear probe as a function of layer depth for different MLP network depths (bump encoding).**

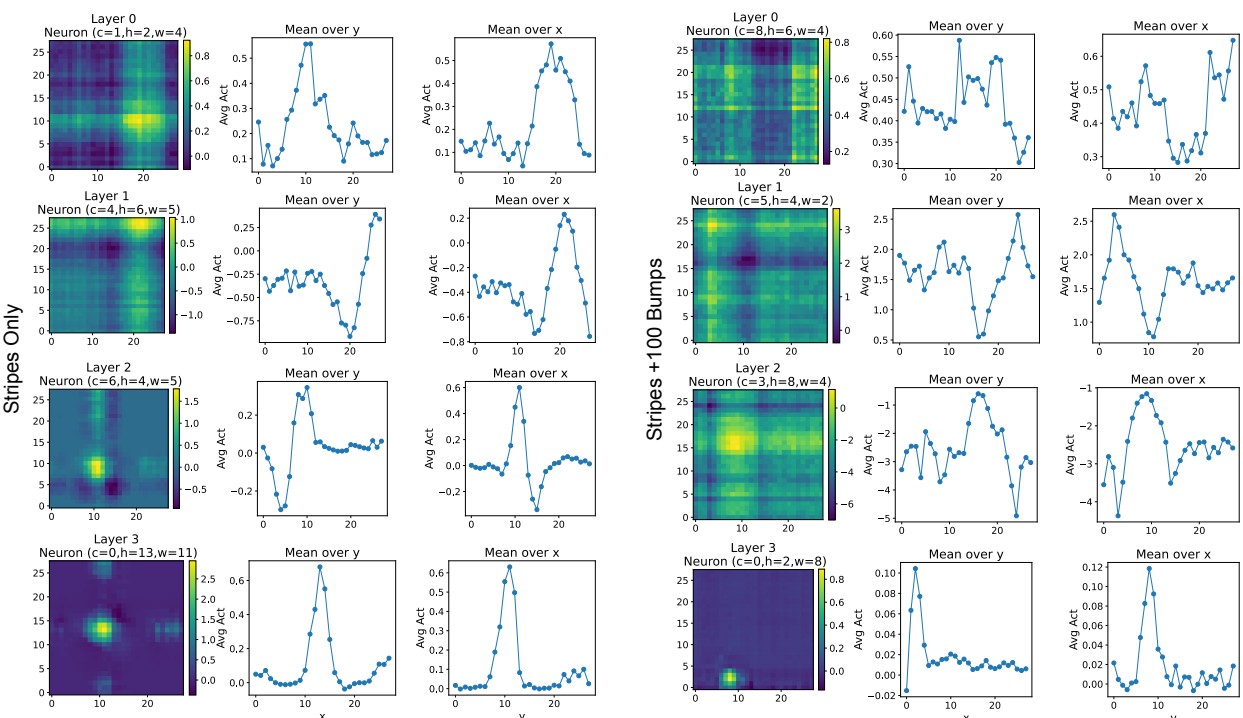

*Figure 15.* **Neural tuning curves for sample neurons across different layers of a 4-layer CNN trained on a dataset of only stripes (left) and stripes + 100 bumps (right).**

