# OpenReview forum: "Compositional Generalization via Forced Rendering of Disentangled Latents"
_ICML.cc/2025/Conference — ICML 2025 poster_

### Official Review · Reviewer_M163 · 2025-03-07

**Overall Recommendation:** 3

**Summary:**

The paper develops theoretical and empirical results as to why a disentangled representation in
input does not necessarily lead to OOD compositional generalization.
First, the paper demonstrates the failure of common generative architectures (decoder-only CNN / MLP) to
perform compositional generalization, even when given disentangled input. This is demonstrated on a task of
generating synthetic images of a 2D Gaussian bump, centered at the (x, y) input coordinates. They then suggest
and test an explanation for this failure - that the input becomes re-entangled in deeper layers. This is shown
quantitatively using kernels and manifold. The paper suggests that instead, the models memorize training data.
Finally, two methods are suggested to improve the compositional generalization: architectural modifications with
regularization, and additional training on curated data.

**Claims And Evidence:**

Strengths:
* Studying the impact of factorization is a very important task, since a breakthrough in compositional embedding can break many limits in current learnable approaches.
* The toy example is analyzed extensively and thoroughly, in terms of memorization, manifold analysis, and augmentation.
* The paper’s demonstration of the memorization of the in-distribution training data is presented clearly and is
justified using the binary kernel factorization. Likewise, the presentation of the re-entanglement using the Jacobian
tensor is valid.
* The low-rank approach is an interesting way to force factorized structure and prevent memorization. In addition,
augmenting the training data allows improving compositional generalization of current models without the need
to modify architecture.


Weaknesses:
* It appears problematic to generalize into holistic and general insights out of a single and limited toy example (it is indeed mentioned in the Limitation section, however this may not be a simple limitation but a biased *general* insight). Experiments may be biased due to various reasons, such as: 1) the relationship between the size of the network and the amount/variability of the learning data may extremely bias the network into memorization instead of generalization. 2) relying on CNN based network for a task containing locality, and hence spatial aware network is preferable, may also be problematic. It can shift the network into memorization, since the architecture is less adequate to solve the problem. For instance, it may be that more complicated networks, specifically spatial aware ones (like transformer containing positional encoding phase) may lead to different conclusions.
* The authors show that subsequent layers re-entangle the representation, however it is not proven to be the cause
of failure for OOD generation. The paper does not test the less constrained hypothesis, that it is sufficient for the
representation to be merely invertible into the original factors. The requirement of strict factorization seems too
constraining.
* The generality and applicability of the suggest methods are unclear without tests on more complicated datasets.
In addition, generating additional curated data is domain-specific and can be very expensive and demanding
for real dataset.
* The paper’s claim - that standard architectures fail to achieve compositional generalization despite being pro-
vided with explicitly factorized inputs — implicitly assumes that a factorized input should be sufficient for such
generalization. However, prior work (e.g. Montero 2021) has already shown that compositional generalization depends on more than
just disentanglement at the input level. Consequently, the paper refutes a weaker claim than what is typically
asserted.

**Essential References Not Discussed:**

Related Work should contain a section on loss terms for compositionality.

**Experimental Designs Or Analyses:**

Insufficient, toy, single architecture that is not optimal for the problem, see above.

**Methods And Evaluation Criteria:**

* Nowadays VAEs fall short in terms of generative capabilities, way beyond diffusion models. I think that theoretically it is important to understand compositional generalization in general, but investigate it on CNNs is reducing the impact of the findings. (Also here I noticed that it mentioned in the Limitation section as well, I agree with this limitation and mention it here as a weakness that it is not "cancel" the insights but tone them down).
* The ablation study is done on various combinations of the loss terms but all on the same architecture. If the tuning is only on the learned loss it can easily be applied to other architectures (which is a pro in fact), then I would expect this type of ablation study.

**Other Comments Or Suggestions:**

see above

**Other Strengths And Weaknesses:**

Minor weaknesses:
* The paper does not list the architectures used for the CNN and MLP used in section 3.1.
* VAE play a central role in generation in general and compositionality research in particular, and hence the paper
is lacking by not including it in its tested architectures.
* The visualizations contain too small font size and information. An extreme example is in (e) in Fig. 4.
* Line 309 right column, a sentence started but not finished, "both results show"
* Not clear that MSE loss for this task is preferable. What if the network required to set a bump in (x,y) and set it in (x+d,y+d). The same MSE loss will be reached regardless the distance from the required. I think that earthmovers metric (optimal transport loss in general) would be more appropriate here.
* Why do the authors decide on this specific task? Why generative task, and under generative ones, why this specific bump task? I did not find intuitive explanation in the paper why this task is preferable for some reason.
* Inconsistencies in capitalizing. For example subsection 3.4 contains capital letter only for the first word, while subsections 4.1, 4.2 and more capitalize each of the words.

**Questions For Authors:**

see above

**Relation To Broader Scientific Literature:**

OK

**Theoretical Claims:**

The methodolgy and reasoning here seems not to take into account all possible cases for their final conclusions. see above.

---

> ### Author Rebuttal · Authors · 2025-03-31
>
> **Architecture Choice (Size & Type)**
>
> - **Data–Model Size and Memorization:** In our experiments, the network does not memorize when we alter the data (e.g., 1D stripes with few conjunctions), even with significantly fewer samples. Thus, we do not see evidence that a “small dataset vs. large model” alone must lead to memorization. Instead, we find that memorization arises when factorization is not encouraged (via e.g. regularization, data augmentation), and it exhibits the “superposition” strategy—activating all relevant seen patterns simultaneously.
>
> - **CNNs/MLPs as Foundational Backbones:** We tested both CNNs and MLPs (the backbone of many generative models, from VAEs, UNet-based Diffusion models, to transformers). Both show manifold distortion and OOD failure, suggesting that “warping” in the presence of incomplete data coverage is fairly universal. We also tried positional encoding with CNN/MLP (omitted for brevity) and observed similar OOD failures.
>
> - **Transformers & Spatially-aware Models:** When we experimented with Transformer-based architectures for the 2D bump task, we found a notable difference: standard Transformer encoder layers with self-attention across pixel tokens performed worse in out-of-distribution bump placements compared to a simpler stack of per-token linear layers and non-linearities. Since each pixel token already has direct access to (x,y) context via positional embeddings, pooling or mixing tokens through self-attention seems unnecessary and can inadvertently entangle factors, degrading OOD generalization. Conversely, a purely feed-forward per-token approach preserves the simple, distance-like mappings needed for forming a 2D bump, yielding better compositional generalization.
>
> **Factorization vs. Mere Reversibility**
>
> - **Why Not Just Invertibility?** We appreciate the suggestion that invertibility might ensure compositional generalization. However, we emphasize compositionality at the manifold level, as single-sample invertibility alone may still allow factors to become entangled across the manifold, hindering new combinations. For instance, invertible models like normalizing flows don't inherently show better compositionality, suggesting that invertibility by itself is insufficient.
> - **Jacobian‐Based Manifold Factorization:** Our Jacobian‐based metric checks how well the manifold locally preserves directions ∂/∂x vs. ∂/∂y If these directions become “cross‐wired,” the network can no longer compose new factor pairs effectively—despite potentially having a globally invertible map. In short, local entanglement disrupts factor reuse.
> - **Partial Rather Than Strict Factorization:** We do not require each neuron or layer to exclusively encode one factor. Rather, we need that the overall manifold maintain approximately independent axes for each factor. Empirically, when the representation heavily mixes them, the model reverts to memorizing ID combinations rather than systematically composing new ones.
>
> Although an invertible transformation can theoretically map individual samples in or out, robust compositional generalization requires more than local invertibility—it requires the activations' distribution to avoid entangling distinct factors, thus enabling reuse in novel combinations. Our Jacobian-based metric tracks this manifold-level factorization, emphasizing sufficient (rather than strict) factorization across the manifold for OOD generalization. We will clarify this distinction to avoid confusion.
>
> **Applicability to More Complex Datasets & Real Data**
>
> See our discussion for Reviewer 3 (iydz).
>
> **Novelty & Positioning Relative to Prior Work**
>
> See our response for Reviewer 3 (iydz).
>
> **Discussion on Loss, Architecture Details, & Additional Points**
>
> - **Loss Functions:** We appreciate the note on Earth‐Mover distance. While it may be more spatially appropriate for “bump” images, MSE is simpler and general. Off‐manifold images, which need not look like bumps, might not benefit from EMD.
> - **CNN/MLP vs. VAEs:** Our “decoder‐only” approach effectively isolates the portion of generative models responsible for mapping factorized latents to pixel space, which also occurs in VAEs and Diffusions. We wanted to remove as many confounding factors as possible to highlight the re‐entanglement phenomenon.
> - **Why This Specific Bump Task?** We wanted a minimal environment with known factorization. This clarity allows us to discover the superposition and warping phenomena in an unambiguous way that might otherwise be obscured in complex tasks.
> - **Minor Stylistic Issues:** We will address all writing and visualization issues raised by the reviewer.
>
> We appreciate the reviewer’s feedback and will clarify this in our revision. Our findings underscore that factorized inputs alone don't guarantee compositional generalization unless factor separation is maintained throughout the network’s forward pass. We hope these insights inform robust model designs and data strategies more broadly.

---

> > ### Comment · Reviewer_M163 · 2025-04-04
> >
> > In light of the rebuttal response to my review and to the other reviewers the contribution is now more clear and I raise my rank.

---

### Official Review · Reviewer_iydz · 2025-03-12

**Overall Recommendation:** 3

**Summary:**

The authors investigate why disentanglement is not sufficient for compositional generalization. First they observe that models are unable to reconstruct simple bumps in unseen locations in visual space from fully factorized latents. They use this as evidence that disentanglement alone is not enough for compositional generalization. They then proceed to show how different forms of regularization and data augmentation can help models to achieve better generalization. These include penalizing the entropy and variance of the singular value of filters in the transformation from representation to input and modifying how data is presented so that models learn about each factor independently.

**Claims And Evidence:**

Yes, the claims made in the article are well supported by empirical simulations.

**Essential References Not Discussed:**

See above.

**Experimental Designs Or Analyses:**

The experimental design and analysis is sound.

**Methods And Evaluation Criteria:**

They do, though as is often the case in ML/DL, there is a hope to see insights in toy datasets translated to more complex datasets. In this case the datasets used are very simple (just gaussian bumps), so it is unclear if these insights translate to other settings.

**Other Comments Or Suggestions:**

I think my main suggestion is to shift the tone of the article from one where the authors claim to make this fundamental discovery (which for better or worse has already been done), to one where they characterize it based on transport perspective. Specifically, they can ask how layer depth and the like affect this issue and then discuss their regularization techniques/data augmentation. I would actually say that this last part feels more like curriculum learning, which has some interesting links to cognitive science and how we learn about concepts in isolation before learning how the interact with each other. Then I would try to translate their insights to more sophisticated datasets (We don't need ImageNet, but there are plenty of datasets that are used to explore disentangled representations out there). Be aware though, this idea of regularization has been tried before and in general does not lead to good results (see references), so the authors need to make a clear case as to why their approach is fundamentally different instead of just mathematically. Finally, I agree that architectural constraints are good, as shown in [4]. The question would be, are the general architectural motifs that apply across modalities/concepts?


[1] Kim, H., & Mnih, A. (2018, July). Disentangling by factorising. In International conference on machine learning (pp. 2649-2658). PMLR.
[2] Burgess, C. P., Higgins, I., Pal, A., Matthey, L., Watters, N., Desjardins, G., & Lerchner, A. (2018). Understanding disentangling in $\beta $-VAE. arXiv preprint arXiv:1804.03599.
[3] Zhu, X., Xu, C., & Tao, D. (2021, July). Commutative lie group vae for disentanglement learning. In International Conference on Machine Learning (pp. 12924-12934). PMLR.
[4] Montero, M. L., Bowers, J. S., & Malhotra, G. (2024). Successes and Limitations of Object-centric Models at Compositional Generalisation. arXiv preprint arXiv:2412.18743.

**Other Strengths And Weaknesses:**

The main strength of the article is that it provides a more thorough account of some findings/insights that have been previously found. Specifically, I believe that viewing compositional generalization failures in generative models from the perspective of transport operators is a valuable one. But since both the findings and insights are not novel it is unclear how useful this will ultimately be.

**Questions For Authors:**

I have no questions apart from the ones above.

**Relation To Broader Scientific Literature:**

They are, but also not very novel. My feeling is that the authors put too much emphasis on what they have found (which is not novel) when they should put more emphasis on their analysis. Specifically, the idea that just having fully disentangled representations does not help compositional generalization was already pointed out in Montero et al., 2021 (which they discuss, making this omission a bit puzzling. See the last section in that article.). Additionally, their design where a central pattern is excluded from reconstruction was already explored in [1], though admittedly it appears in the appendix. Finally, the insight that enforcing some factorization in output space is required to perform compositional generalization was already hypothesized in [2] (final section). Thus the article sits in a weird position where it retreads some points that have already been made, reaching similar conclusions even if they do so via different methods and perspectives (which I believe is still valuable).


- [1] Watters, N., Matthey, L., Burgess, C. P., & Lerchner, A. (2019). Spatial broadcast decoder: A simple architecture for learning disentangled representations in vaes. arXiv preprint arXiv:1901.07017.
- [2] Montero, M., Bowers, J., Ponte Costa, R., Ludwig, C., & Malhotra, G. (2022). Lost in Latent Space: Examining failures of disentangled models at combinatorial generalisation. Advances in Neural Information Processing Systems, 35, 10136-10149.

**Theoretical Claims:**

There were no proofs in the manuscript.

---

> ### Author Rebuttal · Authors · 2025-03-31
>
> **Novelty and Positioning Relative to Prior Work**
>
> We fully agree that claiming the insufficiency of disentanglement is not novel (also see response to Reviewer 1, vKC8). Our primary contribution lies instead in providing a detailed mechanistic explanation of why disentanglement fails—specifically, through manifold warping (due to topological deformation) and superposition-based memorization. We thank the reviewer for highlighting Montero et al. (2022), who primarily attribute compositional failures to encoder limitations rather than decoder issues. In contrast, we focus exclusively on decoder failures. Montero et al. observe that models succeed in non-interactive but struggle with interactive composition, consistent with our findings that the default composition mode is superposition-based, suitable only for non-interactive scenarios. They suggest interactive composition requires learning causal interactions between factors; notably, our proposed methods (embedding regularization and data augmentation) explicitly encourage learning these causal interactions, which we demonstrate to be effective empirically.
> In our revision, we will amend the title and abstract to clearly highlight our contribution as providing deeper diagnostic insights into compositional generalization failures, explicitly acknowledging prior work and positioning our study accordingly.
>
> **Applicability to More Complex Datasets and Real-World Data**
>
> We agree that translating insights from our synthetic study to more complex, real-world datasets is an important future direction. Indeed, as previously emphasized by Montero et al., interactive compositionality cannot be addressed by a one-size-fits-all solution, and accordingly, we do not claim to propose a universal remedy. While our specific approach may not directly generalize to complex datasets, it highlights two valuable exploratory directions: (1) training modular, output-level embedding filters dedicated to each disentangled input dimension, and (2) dataset augmentation with isolated factors of variation. Our simplified setting was specifically chosen to clearly illustrate underlying mechanisms of compositional failure (such as models resorting to memorized ID data for OOD generalization—a nontrivial yet unsuccessful mode).
>
> In our revision, we will explicitly acknowledge that generalizing our diagnostic insights to more complex datasets remains open and valuable. We appreciate the reviewer’s suggestion of explicitly exploring common disentanglement datasets and will discuss this as a key direction for future research.
>
> **Relationship to Curriculum Learning and Cognitive Science**
>
> We greatly appreciate this insightful connection. Indeed, our data curation method—introducing each factor independently—does resonate closely with curriculum learning strategies in cognitive science. In the revised manuscript, we will explicitly acknowledge this parallel, highlighting that isolating concepts before combining them is a recognized effective strategy both in human learning and potentially in artificial neural networks. We will cite relevant cognitive science literature as suggested, discussing this interesting link.
>
> **Regularization Techniques and Architectural Constraints**
>
> We thank the reviewer for this important caution. Our regularization method differs from prior work by explicitly targeting the singular-value structure of the decoder’s weight matrix to mitigate topological deformation (warping). Nevertheless, we recognize and appreciate the reviewer’s caution about prior regularization strategies’ limited success in broader domains. We will clarify this distinction explicitly in our revision, emphasizing that while our targeted regularization approach is effective in our controlled setting, its broader applicability is an open and important question.
>
> Regarding architectural constraints, we strongly agree that identifying general architectural motifs applicable across modalities and concepts is a promising direction. We will explicitly discuss this in the revised manuscript, noting that the identified failure modes and solutions (such as factor-specific architectural constraints) could be generalized across other data modalities.
>
> **Terminology and Clarity**
>
> We thank the reviewer for this valuable suggestion. We will comprehensively revise the manuscript to adopt a more precise and appropriate tone, clearly positioning our paper as providing mechanistic diagnostic insights rather than fundamental novelty in identifying the insufficiency of disentanglement. We agree this will significantly enhance clarity and ensure accurate positioning within the existing literature.
>
> We sincerely thank the reviewer again for their detailed comments, constructive criticisms, and thoughtful suggestions. These revisions will substantially improve the clarity and positioning of our paper, and we appreciate the reviewer’s help in guiding this important improvement.

---

### Official Review · Reviewer_g6yu · 2025-03-13

**Overall Recommendation:** 3

**Summary:**

The paper investigates conditions under which a neural network learns to generalize "compositionally". The setting involves learning to generate a 2D "bump function". A key result is that a "disentangled" representation is not sufficient to ensure compositional generalization. The authors then describe data curation and regularization strategies that appear, in their synthetic setting, to be sufficient for compositionality.

## update after rebuttal: As described in my comment later in the thread, I am keeping my initial rating,

**Claims And Evidence:**

The claims seem reasonably supported, and I appreciated the extensive use of visualizations to analyze the results. The setting is simple enough that the results seem fairly transparent—that's a plus, and kudos to the authors for creating such an easy-to-analyze model.

If I have any reservations, it's that the specific definition of "compositional generalization" (as described briefly in the body of the paper, and more extensively in Appendix A) seems potentially too narrow. When I look at the function learned in Figure 1d, for example, it actually seems like a fairly clever OOD generalization: the network appears to have learned that (1) center pixels are empty, (2) there must be bumps at the given x- and y-coordinates; (3) the centroid of the bumps appears at the given x- and y-coordinates. And this generalization appears "compositional" in the sense that it is superposing (adding) separate solutions for x- and y-coordinates.

**Essential References Not Discussed:**

n/a

**Experimental Designs Or Analyses:**

The experimental designs made sense, but note the question above about experimental detail.

**Methods And Evaluation Criteria:**

The methods seem reasonable for the given toy problem. One issue is that I didn't immediately see exact details on some of the networks used (e.g. for the bump functions). It's not clear to me there are sufficient details to reproduce the authors' experiments precisely.

**Other Comments Or Suggestions:**

I don't think you need to keep adding quotes around the word "superpose".

**Other Strengths And Weaknesses:**

Strengths:
- The results from the initial toy model are not particularly surprising in retrospect, but are a very nice illustration that generalization doesn't always happen in the way that one would expect. I don't know of this example in the literature, and it seems like a good addition.
- The regularization described in section 4 appears to be a simple way to encourage compositional generalization; I'm not 100% sure I understood it, but if I did, it seems potentially useful.

Weaknesses:
- I would have liked a more careful description of why this particular definition of "compositional" was chosen
- Not clear there's enough detail to replicate the experiments exactly.

**Questions For Authors:**

Can you say more about equation 8?

Is code available for this? If so, it would help in reproducing the results; apologies if I missed this in the text.

**Relation To Broader Scientific Literature:**

The related work sections seems good: in particular, I appreciate that it was focused and did not bring in irrelevant details.

**Theoretical Claims:**

Generally the claims made sense to me. However, I have to admit I don't completely follow what's going on with equation 8, and would like to see more detail.

---

> ### Author Rebuttal · Authors · 2025-03-31
>
> We sincerely thank the reviewer for their thoughtful and encouraging feedback on our work. We greatly appreciate the insightful comments and constructive questions raised, and we respond to each point concisely below.
>
> **Definition of Compositionality and OOD Generalization**
>
> We thank the reviewer for this accurate observation. Indeed, compositional generalization, or “OOD generalization” in general is a very broad term. For a task that involves extrapolating far from the training regime (as studied in our work), a central challenge is the lack of well-defined “ground truth.” Thus, we adopted a setting where OOD generalization explicitly involves combining two independently varying factors, representing perhaps the simplest compositional scenario.  Interestingly, even in this minimal setup, our results illustrate that neural networks struggle without additional interventions, highlighting broader challenges in compositional generalization. There are indeed many more different forms of compositionality, see our response to Reviewer 3 (iydz) regarding novelty for a discussion of interactive vs. non-interactive compositionality.
>
> In terms of the model's superposition strategy, while the model indeed learned a highly nontrivial, compositional solution, it is at the activation level, rather than the pixel level, which led to failure of the desired generalization performance, revealing the fact that it is memorizing the ID data rather than breaking them down into composing factors. Our definition of compositionality and our performance metrics aim at assessing the model’s ability to construct novel compositions from factors of independent variation. In the superposition case, the model fails at constructing novel compositions correctly.
>
> **Experimental Details and Reproducibility**
>
> We acknowledge the importance of reproducibility and will provide precise architectural and training details in the revised manuscript. Specifically, we will clarify the architecture of the neural networks used for generating bump functions, including exact layer sizes, activations, optimization algorithms, hyperparameters, and training protocols. Additionally, we will release the code to ensure complete reproducibility and facilitate further exploration of our findings. We have explored a plethora of different input formats and model depths. The architecture details for a sample CNN and an MLP taking bump-encoding inputs are given below:
>
> | Architecture | Input Dimensions          | Output Dimensions | Layers                                           | Hidden Layer Size | Parameter Count (example) | Activation | Optimizer & LR |
> |--------------|---------------------------|-------------------|--------------------------------------------------|-------------------|---------------------------|------------|----------------|
> | **CNN**      | 56 (reshaped to 56×1×1)   | 1×28×28           | ConvTranspose2d (upsampling from 7×7 → 28×28)   | 64 channels       | ~315K (4 hidden layers)   | ReLU       | AdamW (1e-3)   |
> | **MLP**      | 56                        | 1×28×28           | Fully-connected Linear layers                    | 256 units         | ~272K (3 hidden layers)   | ReLU       | AdamW (1e-3)   |
>
> We will include this summary table in the revised manuscript to clearly communicate the architectural details and facilitate reproducibility.
>
> **Clarification of Equation (8)**
>
> We thank the reviewer for pointing out this potential source of confusion. The vectors and eigenvalues referenced in Equation (8) are specifically used to construct the weight matrix within our network. This formulation enables targeted regularization of different components of the weight matrix more efficiently, promoting factorization and preventing representational warping. This is especially helpful since a separate 2D embedding matrix is dedicated to each dimension in the factorized input, which encourages the model to learn for pixel-level factors. Indeed, the low-rank regularization of these embedding matrices encouraged the model to find “stripe-like” factors as shown in Fig. 3b. In the revised manuscript, we will elaborate further on Equation (8), explicitly explaining the mathematical intuition behind this approach and clarifying how it aids compositional generalization.
>
> **Minor comments**
>
> We will revise the manuscript to get rid of quotation marks around superposition accordingly.
>
> In summary, we sincerely thank the reviewer again for these valuable comments, which significantly help enhance our manuscript. We are confident that incorporating these suggestions will improve the clarity, rigor, and accessibility of our paper.

---

> > ### Comment · Reviewer_g6yu · 2025-04-03
> >
> > I appreciate the additional information, which will improve the paper. Note that I am keeping my rating the same, because the main issue, in my view, is the nature of the toy task, and the assumptions about what a "correct" OOD generalization is. As described in my initial review, the solution the network found is justifiable even at the pixel level; this is an ambiguous task. Perhaps for a human, the fact of a bump being "connected" is extremely salient, so a generalization that doesn't preserve connectedness of the output seems somehow "wrong". But there's really no reason to think connectedness should be salient to a learner in this context.

---

> > > ### Author Response · Authors · 2025-04-04
> > >
> > > We sincerely thank the reviewer for the helpful feedback! While we agree that the "connectedness" is a subjective measure of success, it is revealing whether the model has learned the compositional causal models underlying the data. If the model has learned to construct data via learning the correct compositional maps (e.g. composing two 1D gaussians), then it will generalize, meaning constructing ID and OOD data in the same fashion. Our toy task setting is perfect for demonstrating the failure mechanisms because of its simplicity, which would have been otherwise impossible. Indeed as you mentioned that the model does something unexpected and nontrivial when asked to compositionally generalize when it memorizes ID data. Further, we showed that with techniques such as regularization and data augmentation, we can bias the model to learn compositional solutions. Hence, this is why we are convinced that such toy studies are of value for understanding disentanglement and compositionality.

---

### Official Review · Reviewer_vKC8 · 2025-03-13

**Overall Recommendation:** 3

**Summary:**

The authors investigate the role of disentangled representations in compositional generalization, which remains unclear in the literature. The authors observe that while inputs may be disentangled, this disentanglement can "erode" through subsequent layers, such that the model overall is not able to generalize OOD. Looking deeper, the authors find that this "erosion" usually stems from models memorizing training instances. Finally, the "erosion" can be mitigated via architectural constraints or curated datasets. Overall, disentangled representations seem to be a necessary but not sufficient condition for compositional generalization.

**Claims And Evidence:**

1. Factorization alone, independent of input encoding formats, is not sufficient for comp. gen.
    - This claim is mostly well supported by the data, especially Fig. 1b-c.
    - More comprehensive numbers (e.g., MSE ID/OOD for different encodings in a table) would have been nice to see.
    - The different input encodings are not exhaustive, e.g., what about simple (normalized) scalar inputs instead of 1-hot encoding?
    - Does the shape of the ID/OOD region play any role? E.g., if the ID region was a diagonal strip as in [3] (see below), correlation of factors in the training data might be an additional issue.
2. Failures are due to memorization and "superposition" of seen data points.
    - This claim is somewhat supported by the plots in Fig. 1b-d.
    - The kernel perspective from Sec. 3.3 and Fig. 1e should give a more comprehensive perspective, but it is somewhat unclear to me what Fig. 1e depicts, see below.
3. Manifold warping ruins factorization
    - This claim is clear and well supported by the evidence in Fig. 2.
4. Architecture/regularization can encourage compositional generalization.
    - This claim is mostly supported by the visualizations in Fig. 2 and especially the ablation in Fig. 2d.
    - However, the visualizations in Fig. 2b,c are for one specific model only. A more comprehensive comparison, e.g., in terms of average ID/OOD performance over multiple models with/without regularization in a table could give increased certainty that the results hold in general.
5. Dataset augmentations can encourage comp. gen.
    - As with 4, this claim is mostly supported by the results, but more general results averaged over multiple models would increase confidence in the visualizations.

**Essential References Not Discussed:**

- [1] _Montero et al., 2022, Lost in Latent Space: Examining failures of disentangled models at combinatorial generalisation_. This is a follow up to Montero et al., 2021, which the authors cited. It investigates whether failure to generalize compositionally is largely due to a model's inability to disentangle inputs (the encoder), or its inability to generate new compositions (the decoder) and finds decoder errors to be most prominent.
- [2] _Schott et al., 2022, Visual Representation Learning Does Not Generalize Strongly Within the Same Domain_. This paper is similar to Montero et al., 2021, but focuses on the encoder rather than the decoder
- [3] _Wiedemer et al., 2023, Provable Compositional Generalization for Object-Centric Learning_. Similar to [1], this paper shows decoder errors to be responsible for failures to generate compositionally, which can be overcome by additional architectural constraints. The paper discusses how models that disentangle latent factors can be guaranteed to robustly compose them, albeit with a focus on object-centric learning methods.
- [4] _Lachapelle et al., 2023, Additive decoders for latent variables identification and cartesian-product extrapolation_. Like [3], the authors explore compositional generalization in object-centric learning.

Specifically, [1], [3], and [4] hint at the finding that factorization must be actively maintained, e.g., via additional architectural constraints.

**Experimental Designs Or Analyses:**

The analyses are mostly clear, except for:

- Sec. 3.3/Fig. 1e: What does "agreement between the binary factorized kernel and the similarity matrix between the OOD and ID generated samples" mean? It would be helpful if this could be formalized in an equation. What regions of the complete plots in Fig. 8 does Fig. 1e correspond to? What regions in Fig. 1e correspond to ID/OOD? From the y-axis label it seems like the entire plot corresponds to OOD samples?

**Methods And Evaluation Criteria:**

The proposed evaluations make sense for the most part. However, I have some questions about the evaluation setup in Sec. 4.2.:

- Was the fixed coordinate still an input to the network?
- I find the stripes somewhat problematic since, by definition, their superposition results in the Gaussian blob. If the model simply memorizes and superimposes factors as before, this would mostly "solve" the OOD case, as we can see in Fig. 7b with $p=0$%.
- What could this augmentation look like for other types of compositional data? I'm having a hard time imagining an analogy to the 1d stripes for factors such as "color", "shape", "size" in, e.g., a sprites setting.
- How does this compare to model performance when restricting the ID set to x/y combinations with 1 fixed coordinate, but where the output is still a bump (i.e., the dataset consists solely of bumps on the left with varying y, or bumps on the bottom with varying x)? I find this to be there more widely applicable data augmentation that also provides the model with independent influence of each factor.

**Other Comments Or Suggestions:**

- I would suggest picking either "disentangled" _or_ "factorized" and using it consistently throughout
- unclear how the OOD coordinates in Fig. 1e come to be
- Fig. 2a-b, e-f and Fig. 4e are too small to read. There is white space to both sides, a different arrangement of plots might be possible.
- Page 6, left, first paragraph ends in incomplete sentence

**Other Strengths And Weaknesses:**

As outlined above, I believe the main contribution (in its current phrasing) is not novel, however, additional insights provided in Secs. 3.3-4.2 are still interesting.

**Questions For Authors:**

Please refer to the questions above.

**Relation To Broader Scientific Literature:**

In my understanding, the main and titular observation that disentangled representations are not sufficient for comp. gen. is well known in the literature and has been shown in multiple prior works, including Wiedemer et al., 2023 (cited in the paper) as well as [1,3,4] (see below).

Wiedemer et al., 2023, and [3,4] specifically show theoretically that compositional inputs _alone_ are not enough and additional assumptions in the form of regularizations, architectural constraints, or conditions on the training data are required.

That said, the additional kernel and transport perspectives on how disentangled representations might be "diluted" throughout a model, and the (to my understanding) novel regularization scheme in Sec. 4.1. are still interesting, but the main claim should be adapted to properly reflect prior work, which might also have to be reflected in the title.

**Theoretical Claims:**

The paper makes no theoretical claims.

---

> ### Author Rebuttal · Authors · 2025-03-31
>
> **Input Encoding Types**
>
> We appreciate the reviewer's suggestion regarding the comprehensiveness of input encoding formats. Indeed, we have included both rate-based (scalar) and population-based encodings (e.g., 1-hot, Gaussian bumps, and positional encoding), as depicted in Fig. 1. These variations produced qualitatively similar outcomes. We will explicitly clarify this range of encodings and add a table summarizing ID/OOD MSE comparisons for clarity in the revision.
>
> **Shape of the OOD Region**
>
> We agree the shape of the ID/OOD region could influence results. To address this concern, we previously conducted experiments comparing "circular" and "square" OOD regions, which showed no qualitative differences. Regarding correlated factors (e.g., diagonal strips), our primary focus was on the simpler case of independent factors, as even this simpler scenario already presented significant challenges for compositional generalization. We will clarify this reasoning explicitly in our revised manuscript.
>
> **Quantitative Averaging Over Multiple Runs**
>
> We confirm that quantitative results shown in Figures 3 and 4 are indeed averaged over multiple experimental runs. To enhance transparency and reader confidence, we will include a clearly labeled table of these averaged ID/OOD performance metrics in the revised manuscript.
>
> **Fixed Coordinate Inputs**
>
> We appreciate the opportunity to clarify the input setup. The "fixed" coordinate for the stripes was provided explicitly as -1 (off-canvas), ensuring clear separation between stripe and bump datasets. This detail will be clearly stated in the revised manuscript.
>
> **Potential Issues with Stripe Augmentation**
>
> We appreciate the reviewer’s insightful concern about the stripe augmentation potentially promoting superposition-based memorization. The key significance of our stripe augmentation experiment lies precisely in demonstrating compositional generalization: the model generates correct 2D bumps in OOD regions where it never previously observed bumps during training. Even if stripes are memorized individually, the model successfully leverages this information to construct completely novel, unseen bump configurations. Thus, the stripes serve as a spatial scaffold, facilitating pixel-level signals necessary for learning factorized variations within entirely unseen regions. We will highlight this clearly in our manuscript.
>
> **Generalizing Data Augmentation Strategy**
>
> Please refer to our response to Reviewer 3 (iydz) regarding the same point.
>
> **Fixed Coordinate Bumps Experiment**
>
> We appreciate the reviewer’s insightful suggestion. While practically feasible, this approach might not encourage compositional generalization effectively because each Gaussian bump inherently includes both coordinates, regardless of the dataset's internal variation. The key challenge for the model is still establishing the correspondence between different input dimensions and pixel-level output variations. We will explicitly discuss this consideration in our manuscript revision, clarifying the challenge presented by compositional product structures.
>
> **Clarification on Figure 1e and Section 3.3**
>
> We apologize for any ambiguity regarding Fig. 1e. To clarify, Fig. 1e visualizes similarity between representations of OOD-generated samples (sorted by coordinates) and an idealized binary kernel (similarity=1 if coordinates match exactly, 0 otherwise). The "agreement" quantifies how closely the learned similarity matrix aligns with this ideal binary factorized kernel. Specifically, Fig. 1e illustrates overlap between Gaussian bump images centered at coordinates (x,y) and (x′,y’), comparing the ideal binary kernel (top panel) to actual model outputs (bottom panel). Fig. 1e corresponds to the top-left sections within matrices shown in Fig. 8. We will clearly state equations defining "agreement," explicitly describe the data subset (OOD samples), and provide a detailed caption to eliminate confusion.
>
> **Other References**
>
> We thank the reviewer for highlighting important references and will ensure their inclusion and proper discussion in the revised manuscript.
>
> **Regarding Novelty and Main Contribution**
>
> We agree with the reviewer’s insightful point that the observation "disentanglement alone is insufficient for compositional generalization" is not novel. Our primary contribution is identifying and explaining the failure mechanism (superposition and manifold warping), providing mechanistic insights into why factorized latent representations fail to generalize compositionally. Please refer to our response to Reviewer 3 (iydz) regarding the same point. To clearly communicate this, we will revise our manuscript title and abstract accordingly.
>
> **Additional Comments and Suggestions**
>
> We will revise the manuscript accordingly.
>
> Once again, we sincerely thank the reviewer for these valuable suggestions, which will undoubtedly strengthen our paper's clarity and impact.

---

### Decision · Program_Chairs · 2025-05-01

**Decision:**

Accept (poster)

**Comment:**

All reviewers agree that the paper provides insights for better understanding  why disentangled representation does not guarantee compositional generalization, the claims made in the article are well supported by empirical simulations,  and the extensive visualization and analysis help to explain the main points. Although the reviewers raised the concerns on toy examples which might lead to bias, affecting the insights' generalization to other settings, the authors' rebuttal help to address the concerns. Based on the consistent positive reviews and the rebuttal, I recommend weak acceptance.